# Slow presynaptic mechanisms that mediate adaptation in the olfactory pathway of *Drosophila*

## Carlotta Martelli[1,2]*, André Fiala[1]

[1]Molecular Neurobiology of Behavior, Johann-Friedrich-Blumenbach-Institute for Zoology and Anthropology, University of Goettingen, Goettingen, Germany; [2]Department of Biology, Neurobiology, University of Konstanz, Konstanz, Germany

**Abstract** The olfactory system encodes odor stimuli as combinatorial activity of populations of neurons whose response depends on stimulus history. How and on which timescales previous stimuli affect these combinatorial representations remains unclear. We use in vivo optical imaging in *Drosophila* to analyze sensory adaptation at the first synaptic step along the olfactory pathway. We show that calcium signals in the axon terminals of olfactory receptor neurons (ORNs) do not follow the same adaptive properties as the firing activity measured at the antenna. While ORNs calcium responses are sustained on long timescales, calcium signals in the postsynaptic projection neurons (PNs) adapt within tens of seconds. We propose that this slow component of the postsynaptic response is mediated by a slow presynaptic depression of vesicle release and enables the combinatorial population activity of PNs to adjust to the mean and variance of fluctuating odor stimuli.

DOI: https://doi.org/10.7554/eLife.43735.001

## Introduction

Internal sensory representations can carry information about both the identity of an object and its location in space. This is also the case for odor signals, that are used as cues for both source identification and localization. Whereas a brief exposure to the stimulus can be sufficient for an animal to identify the odor (*Bhandawat et al., 2010*; *Rinberg et al., 2006*; *Szyszka et al., 2012*), changes in stimulus intensity or in higher order statistics, such as variance or temporal correlations, can be used to infer stimulus location (*Baker et al., 2018*; *Celani et al., 2014*; *Murlis et al., 2000*; *Vergassola et al., 2007*). One key challenge of sensory systems is to keep the object identity invariantly represented in terms of neural activity as the animal moves around, the object moves, or the overall environment changes. This is not a trivial problem for two reasons. First, sensory stimuli are typically multidimensional and their features are represented by combinations of neurons with different receptive fields. Therefore, the same stimulus might be represented by different subsets of neurons at different times/locations. Second, the sensitivity of a single neuron is usually not sufficient to encode the full range of natural stimuli, and intracellular or circuit mechanisms may adapt its dynamic range on different timescales to match the concurrent stimulus statistics (*Wark et al., 2007*). In olfaction adaptation to localization cues (as stimulus intensity and variance) could potentially compromise the encoding of identification cues (combinatorial activity).

Here, we use the olfactory system of the fruit fly *Drosophila melanogaster*, a key model for studying olfactory circuits (*Galizia, 2014*; *Su et al., 2009*; *Vosshall and Stocker, 2007*; *Wilson, 2013*) to investigate how adaptive features of distinct neuron types modify the stimulus representation at the population level. Odor stimuli activate olfactory receptor neurons (ORNs) that usually express a single chemosensory receptor out of a large family of genes (*Su et al., 2009*; *Vosshall and Stocker,*

*For correspondence:
carlotta.martelli@uni-konstanz.de

Competing interests: The authors declare that no competing interests exist.

*2007*). Each receptor has different affinities to different odorants, such that a given odor will drive a specific activity pattern across the entire population of ORNs (*de Bruyne et al., 2001*). ORNs expressing the same receptor innervate the same glomerulus of the antennal lobe (AL) (*Couto et al., 2005*; *Fishilevich and Vosshall, 2005*), within which they synapse onto mostly uniglomerular projection neurons (PNs, *Figure 1a*) (*Marin et al., 2002*; *Wong et al., 2002*). In the AL, odor stimuli induce specific patterns of activity across glomeruli (*Fiala et al., 2002*; *Wang et al., 2003*). The PN responses are shaped by properties of ORN-to-PN synapses, such as short-term presynaptic depression (*Kazama and Wilson, 2008*) and postsynaptic receptor dynamics (*Nagel et al., 2015*). Moreover, a network of local interneurons (LNs) reorganizes the responses of the glomeruli based on the overall activity of the AL by means of both excitatory (*Olsen et al., 2007*; *Yaksi and Wilson, 2010*) and inhibitory lateral connection (*Olsen and Wilson, 2008*; *Root et al., 2008*; *Silbering and Galizia, 2007*).

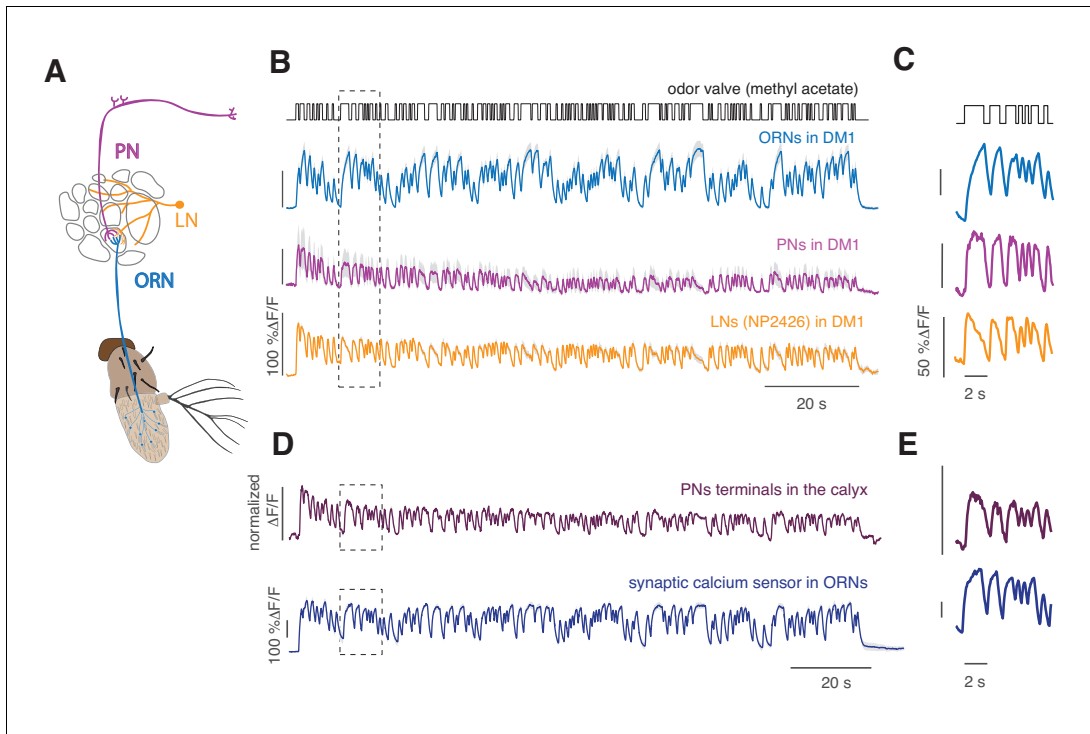

**Figure 1.** Cell-specific calcium dynamics in neurons of the antennal lobe. (A) Schematics of the olfactory pathway in *Drosophila*. ORNs expressing the same receptor project their axons from the antenna into the same glomerulus in the antennal lobe. ORNs synapse onto uniglomerular PNs that send their axons into the calix of the mushroom body (MB) and the lateral horn (LH). (B) *From top to bottom*: open-closed state of the odor delivery valve. The stimulus consisted of a sequence of odor pulses and gaps with random durations between 300 ms and 2.7 s. Calcium responses to a 2 min long random stimulus (methyl acetate $10^{-6}$) measured within the same glomerulus DM1 in ORNs (orco-GAL4), PNs (GH146-GAL4) and LNs (NP2426) (shaded area indicates SEM, *n* = 6–7). (C) Same as in the rectangle in B. (D) *Top*: Calcium response from the axon terminals of PNs expressing the cytosolic calcium reporter GCaMP3 (methyl acetate at concentration $10^{-4.3}$). Several bouton-like regions of interest (ROIs) were selected in each animal (three animals, 35 ROIs), the relative change in fluorescence was calculated for each ROI and then normalized by the maximum value. Response was then averaged across all ROIs. Shaded area (barely visible) represents SEM. *Bottom*: response of glomerulus DM1 reported by a synaptically tagged calcium reporter Syp-GCaMP expressed in ORNs. (E) Same as in the rectangles in (D).

DOI: https://doi.org/10.7554/eLife.43735.002

The following figure supplements are available for figure 1:

**Figure supplement 1.** Odor Stimulus.

DOI: https://doi.org/10.7554/eLife.43735.003

**Figure supplement 2.** Response to sustained fluctuating stimuli reported by GCaMP6f.

DOI: https://doi.org/10.7554/eLife.43735.004

Many studies have characterized adaptation in the olfactory system based on the electrophysio-logical analysis of individual olfactory neurons (*Ito et al., 2009*; *Kaissling et al., 1987*; *Kurahashi and Menini, 1997*; *Lemon and Getz, 1997*; *Nagel and Wilson, 2011*). In *Drosophila*, ORN firing rates adapt to sustained odor stimuli on timescales on the order of ~100 ms (*Gorur-Shandilya et al., 2017*; *Martelli et al., 2013*). This adaptation shifts the dynamic range of the ORN response to match the mean and variance of the stimulus (*Cafaro, 2016*; *Gorur-Shandilya et al., 2017*). However, it is less clear how these adaptive firing rates convert into glomerular activity of populations of neurons downstream of the antenna and further in second-order neurons within the brain. Previous studies in locusts (*Geffen et al., 2009*; *Stopfer et al., 2003*), moths (*Jacob et al., 2017*) and rats (*Gupta et al., 2015*) have reported complex and diverse response dynamics in second-order olfactory neurons. In *Drosophila*, short-term depression, different receptor currents, and lateral inhibition have been identified as mechanisms underlying complex PN response dynamics (*Nagel et al., 2015*). However, it remains unclear how much of the PN dynamics is inherited from ORNs (*Cafaro, 2016*; *Kim et al., 2015*), whether additional adaptation occurs in the AL, on what timescales and how it affects the combinatorial odor representation (*Huston et al., 2015*).

Here, we report that calcium responses at the axon terminals of the ORNs adapt only marginally to repeated or sustained odor stimulation, in contrast with properties of firing activity previously measured at the antenna. Our analysis of pre- and postsynaptic calcium activity is consistent with a model that describes most of the adaptation observed in PNs to occur at the ORN-PN synapses. We identify a slow change in PN activity that occurs on a timescale of 10–20 s and correlates with depression of vesicle release at the ORN terminals. As a consequence of these different properties, the combinatorial representation of an odor in ORNs does not change over prolonged repeated stimulations, while the combinatorial representation in PNs adjusts over a timescale of tens of seconds. We show that this adjustment preserves information about the odor stimulus and allows the PNs to encode stimulus fluctuations on a larger range of intensities than the population of ORNs.

## Results

### Cell-specific calcium dynamics in the antennal lobe

Adaptation of the responses of ORNs to odor stimuli has been reported in several insect species (*Ito et al., 2009*; *Kaissling et al., 1987*; *Lemon and Getz, 1997*; *Nagel and Wilson, 2011*) and in vertebrates (*Kurahashi and Menini, 1997*). In *Drosophila*, ORNs show phasic firing rate responses on timescales of ~100 ms when stimulated with a step increase in odor and adapt their gain to the mean stimulus (*Gorur-Shandilya et al., 2017*; *Martelli et al., 2013*). Here, we used calcium imaging to investigate adaptation properties downstream of the receptor site, that is in the glomeruli. Following an approach previously adopted to quantify firing rates of sensory neurons (*Chichilnisky, 2001*), including olfactory neurons (*Geffen et al., 2009*; *Martelli et al., 2013*), we designed a pseudorandom stimulus (*Figure 1—figure supplement 1*) that allowed us to quantify the dynamics of calcium response. From previous analysis of ORNs spiking activity, we expected that calcium signals measured at the ORN axon terminals would decrease over time when a sustained flickering stimulus is applied (*Martelli et al., 2013*). Instead calcium response, reported by GCaMP3 (*Tian et al., 2009*) or GCaMP6f (*Chen et al., 2013*) (*Figure 1—figure supplement 2*), is sustained, as shown for the ORNs that innervate glomerulus DM1 in response to methyl acetate ($10^{-6}$ dilution, *Figure 1A–B*, cyan). On the contrary, PNs and LNs that innervate the same glomerulus show a decrease in calcium response over time (*Figure 1A–B*). The calcium dynamics of ORNs, PNs and LNs are different also on shorter timescales (*Figure 1C*). Calcium onset is faster in PNs than ORNs, as previously shown in measurements of the firing rate (*Bhandawat et al., 2007*), and LNs show even more transient dynamics with a larger decrease in activity during a single odor pulse (*Nagel and Wilson, 2016*). This demonstrates that the fluorescence calcium reporter, although slower than an electrophysiological measurement, can report cell-specific response dynamics.

Calcium dynamics in ORNs reported by the cytosolic calcium sensor CGaMP3 are similar to the dynamics of the presynaptically targeted calcium sensor Syp-GCaMP3 (*Pech et al., 2015*) (*Figure 1D–E*) and considerably different from the calcium signals measured from synaptic boutons of PNs in the calyx of the mushroom body (*Figure 1D–E*). These results indicate that the calcium dynamics characterized in ORNs terminals reflect cell-specific synaptic activity.

## Dynamic properties of ORN calcium responses to binary stimuli

One common approach to characterize neural response to stimuli is that of using reverse correlation (*Chichilnisky, 2001*) to extract the linear filter of the response. The linear filter describes how the response depends on current and past stimuli in a system that is not an instantaneous translator of the stimulus into a response but has a 'memory' for previous stimulations. This approach can be applied to calcium responses to capture cell specific dynamics (*Si et al., 2019*). For example, different linear filters predict the calcium transients elicited by the random stimulus in ORNs and PNs of the glomerulus DM1 (*Figure 2A* and *Figure 7—figure supplement 1*). In ORNs, the linear filter is monophasic and can be accurately described by a single exponential function. In PNs, the filter is biphasic with a lagging negative component that indicates adaptation, in the sense that stimuli in the past suppress calcium signals in response to present stimuli. These dynamics are certainly cell specific, but cannot be directly interpreted as calcium dynamics in absolute terms, as they could be affected by and depend on the kinetics of the calcium reporter. For example, when using the faster fluorescence sensor GCaMP6f, the reverse correlation analysis results also in a monophasic filter, but with a faster timescale (~100 ms, *Figure 2B*). This suggests that the real presynaptic calcium dynamics are likely faster than the sensor kinetics. Following the approach of *Schnell et al. (2014)*, we have used an exponential filter to model the sensor kinetics and deconvolved it from the ORN calcium response reported by two different sensors (*Figure 2C*). The deconvolved signal shows fast transients, in accordance with the phasic firing rates of these neurons. The deconvolution does not affect the response on longer timescales, which remains sustained for the duration of the 2 min stimulation. This approach does not take into account possible non-linearities of the sensors, which could hide other features of the real calcium response. Importantly, we assume the same model for the sensor kinetics in different neuron types. This implies that cell- as well as stimulus-specific differences reported by a given sensor are preserved through the deconvolution of the sensor kinetics (see below).

Adaptation to different stimuli could in principle change the ORN response kinetics, the response gain or both. Therefore, we tested the response in glomerulus DM1 to six concentrations of methyl acetate (*Figure 2D*) and compared response dynamics as well as changes in gain or other non-linearities of the response. Odor-evoked responses to higher stimulus concentrations can be fitted with similar monophasic linear filters (*Figure 2E*). Linear filters can also be extracted after deconvolution of the sensor kinetics. In this case, they are faster and with a small negative lobe (*Figure 2I*) but are consistently similar at all concentrations.

In order to quantify possible non-linearities in the response, we plotted the measured calcium signals against the convolution of the linear filter with the stimulus (See Materials and methods, *Figure 2F*, gray dots). This relationship could be fitted by a linear function at low concentrations and by a sigmoidal function at higher concentrations (*Figure 2F*, colored lines). This so-called Linear Non-linear (LN) model (linear filter and static non-linear function) (*Chichilnisky, 2001*) predicts well the calcium response of the glomerulus at all concentrations (*Figure 2D*, colored lines). Interestingly, the slope of the static function decreases with higher stimulus concentrations (note log scale in the x-axis), indicating a decreased response gain. We asked whether this different gain is a result of adaptation to the different concentrations or whether it reflects a front-end non-linearity in the receptor response. It has been shown before that the response to stimuli that shut to zero (plume-like or binary stimuli) are indeed affected by the receptor non-linearity (*Gorur-Shandilya et al., 2017*). This is due to the fact that the neuron firing rate in response to isolated odor pulses does not grow linearly with stimulus concentration, but rather shows a sigmoidal dose response curve (*de Bruyne et al., 2001*). Therefore, we estimated a front-end non-linearity by fitting a Hill function to the mean peak response estimated at each stimulus intensity (*Figure 2G*). When using this non-linearity in front of the LN model (see Materials and methods, NLN model), the static functions overlap onto a similar shape (*Figure 2H*), also when using the sensor-deconvolved calcium response (*Figure 2J*). The response gain shows only a small dependency on stimulus mean (*Figure 2J*, inset), demonstrating that a single model of the calcium signals is sufficient to describe the neurons' responses at all stimulus intensities. These results suggest that, apart from a fast transient in the response, little stimulus-driven adaptation occurs in the pre-synaptic calcium at the ORN axon terminals in response to binary stimuli when front-end non-linearities are taken in account.

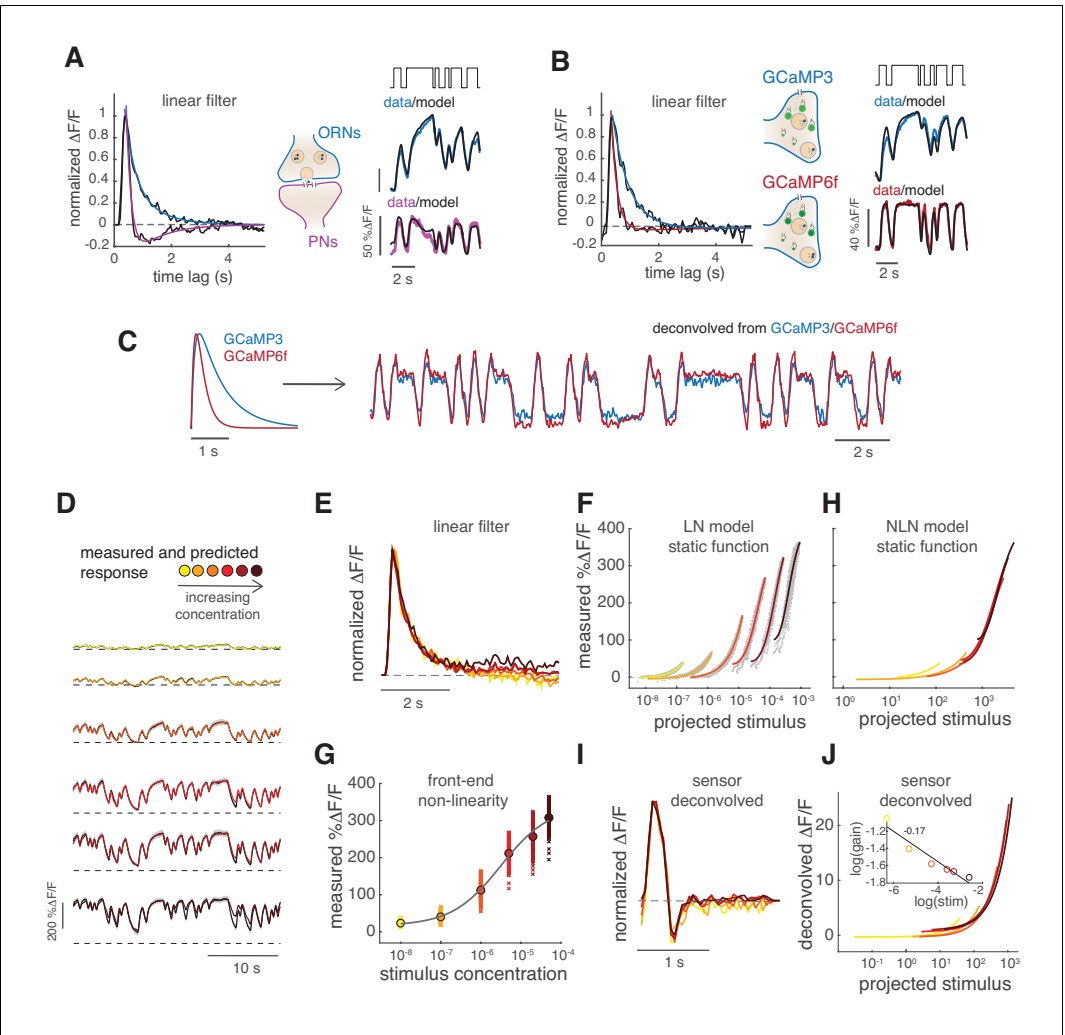

**Figure 2.** Calcium dynamics in ORNs axon terminals. (**A**) Linear filter (black) of the DM1 ORNs and PNs response (methyl acetate at $C_{04}$, $n$ = 11) and fitted exponential decay (cyan $k_1(t) = A_1 e^{-\frac{t}{\tau_1}}$) or double exponential (purple: $k_3(t) = A_1 e^{-\frac{t}{\tau_1}} + A_2 e^{-\frac{t}{\tau_2}}$). *Left:* measured response (black) and response predicted by a LN model (cyan/purple) for DM1 ORNs and PNs. Goodness of fit was quantified as the ratio between the mean squared residual and noise (see Materials and methods and *Figure 7—figure supplement 1*), here NR=0.12. (**B**) Linear filters (black) and fitted exponential decay (cyan/red) for the response of DM1 ORNs expressing either GCaMP3 (cyan) or GCamP6f (red). *Left:* mean measured response (black, $n$ = 6-8) and response predicted by a LN model (cyan/red, NR=0.29/0.48). (**C**) Exponential filters used to model the dynamics of the sensor kinetics ($\tau$ = 0.7s for GCaMP6f and $\tau$ = 0.2s for GCaMP6f) and ORN calcium response deconvolved from data in (**B**). Results are robust to variations in the exact value of the timescale. (**D**) Measured (black) and predicted (colored) response for ORNs in DM1 at different concentrations of the random stimulus, increasing from yellow to dark red ($n$ = 9–11). (**E**) Linear filters fitted at increasing odorant concentrations. (**F**) Static functions (colored) resulting from either a linear or sigmoidal fit to the instantaneous measured response as a function of the projection of the stimulus on the linear filter (gray dots, see Materials and methods). (**G**) Estimate of the front-end non-linearity. Crosses: peak response to each pulse in the random series for the seven stimulus concentrations. Circle: mean peak response. Gray line: Hill function fitted to the mean peak response. (**H**) Static functions resulting from a NLN model of the calcium response. These functions were obtained as in (**F**) but passing the stimulus first through the front-end non-linearity estimated in (**G**). (**I**) Linear filters calculated from the calcium signal after deconvolution of the sensor kinetics and (**J**) corresponding static function. *Inset:* logarithm of response gain quantified as the slope of the static function and plotted as a function of the logarithm of the concentration. The line indicates a linear fit and the number is the slop of the linear function.

DOI: https://doi.org/10.7554/eLife.43735.005

## Background adaptation in ORN firing rate and presynaptic calcium response

It has been previously shown that the response sensitivity of ORNs decreases when they have adapted to an odor background (*Martelli et al., 2013*) and that the ORN firing rate follows the Weber-Fechner law (*Cafaro, 2016*; *Gorur-Shandilya et al., 2017*). If calcium dynamics in the ORNs axon terminals were just a proxy for the neurons' firing activity, then similar properties should be observed in the calcium responses. Therefore, we measured how firing rate and calcium are affected by adaptation to an odor background (*Figure 3A*). Single sensillum recordings (SSR) from ab2A show that the firing rate elicited by a pulse of methyl acetate is lower when the pulse is presented on a background of the same odor (*Figure 3B,D,E*). On the contrary, the calcium response from the axon terminals of the same ORN type (innervating DM4) is not decreased by an adapting background; rather it is slightly increased (*Figure 3C,D,E*). Similar results were obtained with GCaMP6f (*Figure 3—figure supplement 1*). To demonstrate that this property of the calcium dynamics cannot be explained by the sensor kinetics, we have convolved the firing rate with a linear filter modeling the kinetics of the sensors (*Schnell et al., 2014*)(*Figure 3F*). For both GCaMP3 and GCaMP6f, the model predicts a decrease in activity during background presentation and a lower response to the odor pulse. Rather, the actual calcium response is sustained on long timescales and remains always approximately constant (or slightly higher) in response to a pulse independent of background concentration (*Figure 3G*). We could conclude that calcium in axonal terminals encodes the actual stimulus concentration rather than the change in concentration. However, this is not the case when the test pulse is lower than the background: in this case the calcium response becomes lower than in the non-adapted condition (*Figure 3G*, orange). To make sure that in our experiment we did not saturate the receptor response, the calcium signal or the calcium sensor, we have measured both the firing rate and the calcium response for increasing concentrations of the odor (*Figure 3H*). We found that the same stimuli elicit a similar range of responses in firing rate and presynaptic calcium. Firing rate adaptation occurs for all tested pulse concentrations, even for those close to saturation (*Figure 3I*, top). On the contrary, glomerular calcium responses do not decrease upon adaptation, unless the adapting background is higher than the test pulse (*Figure 3I*, bottom). Similar results were obtained for glomerulus DM1 (ab1B) and using the calcium reporter GCaMP6f (*Figure 3—figure supplement 1*). It seems unlikely that non-linearities in the sensor kinetics can explain these results. On one end, saturation of the sensor would not allow the increase in fluorescence observed for the pulses on a background. On the other end, it could be that when the basal calcium is elevated the sensor kinetics enter a regime of larger gain. However, similar results were obtained also for responses in the low range of the firing-to-calcium calibration curve (*Figure 3H*), suggesting that the calcium sensor is in a linear regime of responsiveness. We conclude that glomerular calcium responses do not reflect adaptation observed in firing rate and encode increases and decreases in concentration differently.

## Slow adaptation in odor-evoked calcium activity of PNs

We next asked how sustained but fluctuating stimuli are encoded in the calcium responses of PNs. We have noted already that whereas the ORNs followed the odorant dynamics in a sustained way, the responses of PNs slowly decreased (*Figure 1B*). It reproducibly took approximately 20 s until the calcium dynamics reached a steady state. A slow change in the electrophysiological activity of PNs has been observed upon the onset of a flickering stimulus in flies (*Nagel et al., 2015*) and moths (*Jacob et al., 2017*). We quantified this slow decrease by taking the peak response to individual pulses in the pseudorandom sequence and fitting an exponential function to it (*Figure 4A-B*). The PN responses in DM1 decreased to ~50% of the initial response (*Figure 4C*, purple). The decay timescales extracted by the exponential fit ranged between 3 and 10 s for DM1 and showed no dependency on the initial response (*Figure 4D*). The activity recovered from this slow adaptation within a minute (*Figure 4—figure supplement 1*) and it is therefore of a different nature than what was previously observed in the locust (*Stopfer and Laurent, 1999*) or in the fly (*Das et al., 2011*). The same analysis was conducted for nine other glomeruli (*Figure 4E*). In all cases, ORN responses were sustained for 2 min after stimulus onset, with a mean degree of adaptation across glomeruli close to zero (*Figure 4F*). By contrast, the PN responses in all glomeruli were adaptive, with a mean degree of adaptation across glomeruli equal to 0.57 (*Figure 4F*). The adaptation timescale had no clear

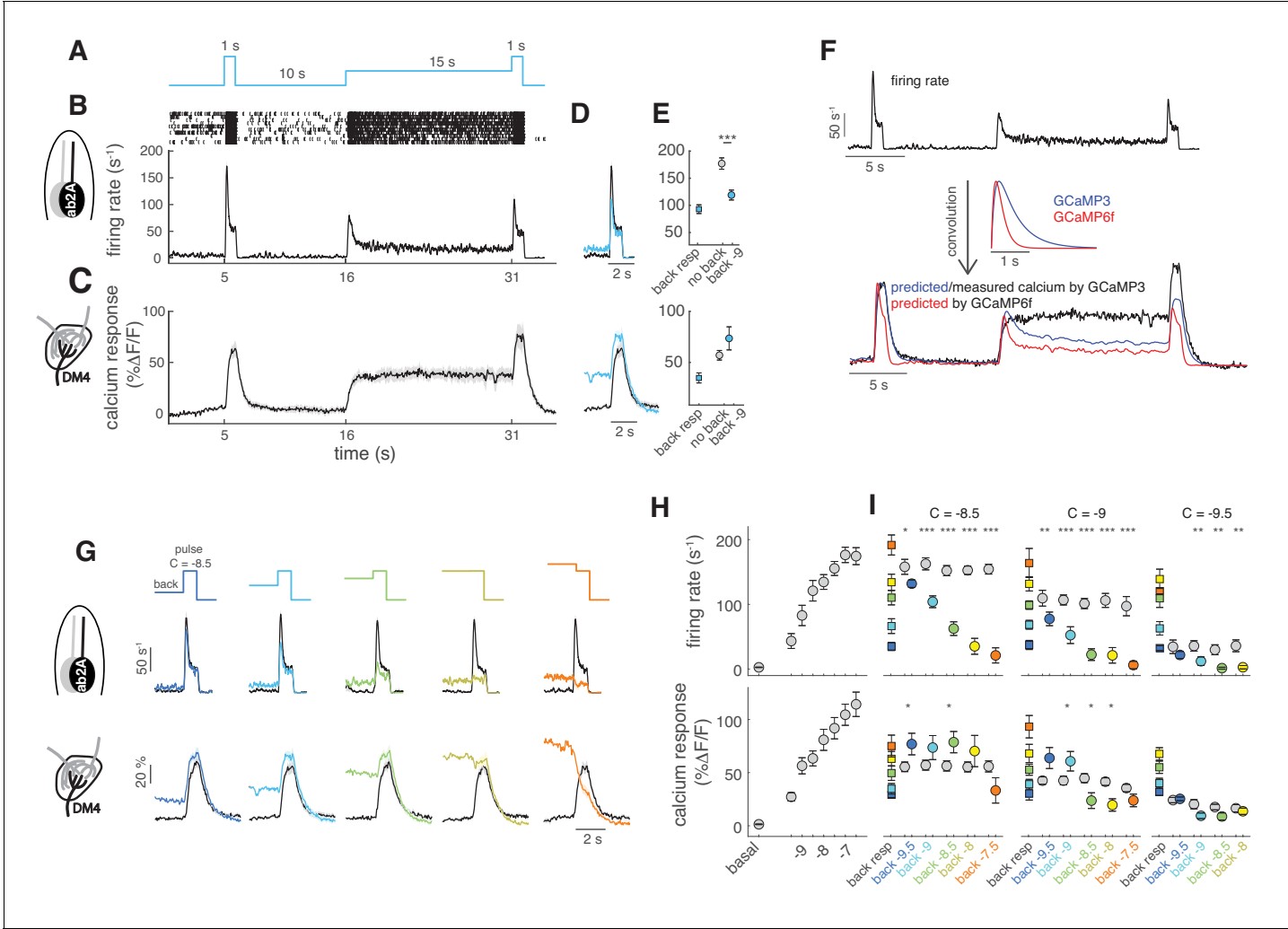

**Figure 3.** Background adaptation in ORN firing rate and calcium responses. (**A**) Stimulus protocol: a pulse of methyl acetate ($10^{-8.5}$ dilution) was presented first isolated and on top of a background of the same odor at lower concentration ($10^{-9}$) delivered for 15 s. (**B**) Raster plot and mean firing rate for the spiking response of ab2A (expressing OR59b) measured in single sensillum recordings, n = 10. (**C**) Mean calcium response measured at the axon terminals in the corresponding glomerulus DM4, n = 8. (**D**) Overlay of the response to the isolated odor pulse (black) and the pulse presented on the background (cyan) for firing rate (top) and calcium (bottom). (**E**) Peak response to background stimulation (squares), to the isolated pulse (gray circle) and to the pulse on background (cyan circle). Paired ttest, ***p<0.001, n = 10. (**F**) Convolution (bottom traces) of the firing rate (top trace) with linear filters representing GCaMP3 and GCaMP6f kinetics. (**G**) Overlay of the response to the isolated odor pulse (black) and the pulse presented on the background (colored) for five increasing values of the background concentration, n = 8–10. (**H**) Dose-response curve measured as mean firing rate (n = 6–11) and mean calcium response (n = 10–11) for increasing concentrations of the isolated odor pulse. (**I**) Mean response to the different odor backgrounds (squares), to the isolated pulse (gray circles) and to the pulse on background (colored circles) for three pulse concentrations (reported in log scale). Paired ttest, *p<0.05, **p<0.01, ***p<0.001, n = 7–10 for firing rate and n = 6–8 for calcium responses. Shaded areas and error bars indicate SEM. Calcium responses were measured in the same flies for figure (**H**) and (**I**).

DOI: https://doi.org/10.7554/eLife.43735.006

The following figure supplement is available for figure 3:

**Figure supplement 1.** ORN background adaptation.

DOI: https://doi.org/10.7554/eLife.43735.007

relation to the amount of initial activity of the glomerulus (*Figure 4G*). For example, responses of high amplitude adapted slowly in glomerulus D and DC1 ($\tau \sim 30$ s), but fast in DL1 and DL5 ($\tau \sim 4$ s). Overall, we conclude from this analysis that calcium activity in PNs decreases in all glomeruli proportionally to the initial response on timescales that vary in the range between 1 and 40 s, and is glomerulus and concentration dependent.

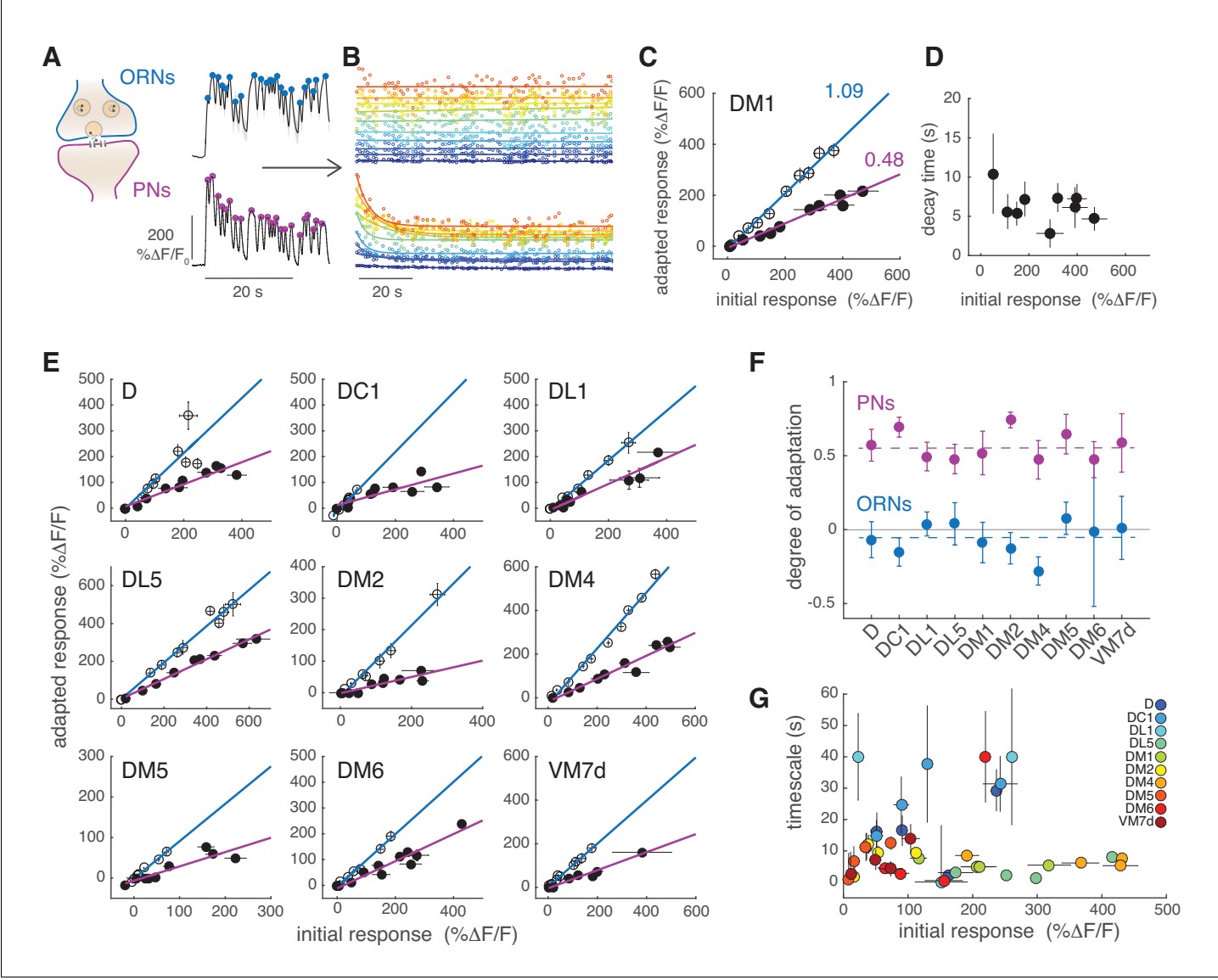

**Figure 4.** PNs adapt on slow timescales to a sustained, fluctuating odor stimulus. (**A**) Initial calcium response reported by GCaMP3 to the first 30 s of random odor stimulation in ORNs and PNs within glomerulus DM1. Dots indicate the peak response to each single odor pulse in the random stimulus sequence calculated from the mean $\Delta F/F$ (n = 11, 7). (**B**) All trials for glomerulus DM1 were pooled, sorted by amplitude and averaged in 10 bins spanning the entire response range (see Materials and methods for detail). Peak amplitude of the mean response to each single random pulse is plotted as a function of time and color-coded by the amplitudes for ORNs (top) and PNs (bottom). Continuous lines represent linear (ORNs, $r_m(t) = A + Bt$) or exponential (PNs, $r_m(t) = A + Be^{-\frac{t}{\tau}}$) fits to the time-dependent peak responses. (**C**) Adapted response as a function of the initial response for ORNs (empty circles) and for PNs (filled circles). Initial and adapted responses were calculated using the fitted parameters. Error bars indicate 95% confidence intervals. Continuous lines represent a linear fit. Numbers indicate slopes of the linear relationship. (**D**) Adaptation time constants of PN responses for different response amplitudes estimated from the exponential fit. Error bars indicate 95% confidence intervals. (**E**) Adapted response as a function of the initial response for nine glomeruli and corresponding linear fit. (**F**) Degree of adaptation estimated as 1 minus the slope of the linear relationship between the adapted and initial response. Dashed lines indicate the mean degree of adaptation across glomeruli: -0.06 ± 0.1 for ORNs and 0.57 ± 0.1 for PNs (mean ± standard deviation). The gray continuous line indicates no adaptation. Error bars indicate 95% confidence intervals (n = 5–11). For all glomeruli, the degree of adaptation is significantly different from zero (p<0.001). (**G**) Adaptation timescales estimated for four odorant concentrations, color-coded by glomerulus, and plotted as a function of response amplitude. Error bars indicate 95% confidence intervals (n = 5–11). See also *Figure 4—figure supplement 1*. Note that a slow decay in activity that outlasted the stimulus duration was observed at the lowest and highest concentrations in DL1 and at the highest concentration in DM6. For these measurements we report confidence intervals for an exponential fit with the maximum timescale (which was set to 40 s in our analysis).

DOI: https://doi.org/10.7554/eLife.43735.008

The following figure supplement is available for figure 4:

*Figure 4 continued on next page*

*Figure 4 continued*

**Figure supplement 1.** PNs activity recovers from slow adaption within a minute.

DOI: https://doi.org/10.7554/eLife.43735.009

## The role of lateral inhibition in PN slow adaptation

We next asked which mechanisms mediate the slow adaptation in PN responses to odor stimuli. One possibility is that the network of inhibitory, GABAergic LNs causes adaptation, for example via GABAergic chloride channels (*Nagel et al., 2015*; *Silbering and Galizia, 2007*). Therefore, we tested whether the PN response dynamics are affected by the application of the GABA-A receptor antagonist picrotoxin (PTX). Application of PTX increased ORN responses in two glomeruli (*Figure 5A–B*), and induced a small but significant change in the response amplitude of PNs (*Figure 5D* and *Figure 5—figure supplement 1*). However, the degree and timescale of adaptation were similar before and after PTX application (*Figure 5D*). We conclude that GABA-A receptor activity does not significantly affect PN dynamics on long timescales and cannot solely account for the slow adaptation observed.

It has been shown that GABA-B receptors in ORNs mediate presynaptic inhibition in ORN terminals in the AL (*Olsen and Wilson, 2008*; *Root et al., 2008*). The effect of this inhibition should already be observable in the presynaptic calcium signals that we measured in ORNs. However, since the presynaptic calcium does not adapt, it makes it an unlikely mechanism. To rule this out, we expressed a previously described GABA-B receptor RNAi construct (*Root et al., 2008*) in ORNs and measured responses from PNs (*Figure 5E*). This manipulation affected the initial response of DM4 PNs (*Figures 5E-F* and *Figure 5—figure supplement 1*), but not the degree and timescale of adaptation (*Figure 5F*). It is possible that the RNAi has an incomplete effect. Therefore, we have tested the GABA-B antagonist CGP54626 and the agonist SKF97541, which have been shown to increase or decrease the calcium response in ORNs axon terminals, respectively (*Root et al., 2008*). CGP54626 application increased the response of glomerulus VM2 to ethyl butyrate and slightly increased the response of DM4, but did not affect DM1 response to methyl acetate (*Figure 5G*), in agreement with the results obtained with the RNAi line. Accordingly, the response of both DM1 ORNs and PNs remained unchanged after CGP54626 application. Therefore, even though GABA-B mediate inhibition can affect presynaptic activity, the timescales and degree of the stimulus driven depression are not directly controlled by GABA-B (*Figure 5I*). Treatment with the GABA-B agonist SKF97541 decreased the ORN response in all three glomeruli measured (*Figure 5H*), demonstrating that also DM1 response can be reduced via GABA-B. Interestingly, tonic inhibition of the GABA-B receptors strongly affected the response to a fluctuating stimulus. In DM1 ORNs, the response to the first odor pulse was lower after application of SKF97541 (*Figure 5J*, top), similarly to the response to the isolated puff, but it increased over the following odor pulses. In PNs, on the contrary, the response slowly decreased and almost disappeared (*Figure 5J*, bottom). This suggests that the effect of lateral inhibition is activity dependent and tightly tuned to the stimulus input through complex interactions in the AL network, as also shown in *Nagel et al. (2015)*. Since the PNs' slow adaptation was observed also in conditions of reduced lateral input, we conclude that GABA-mediated lateral inhibition does not constitute the main mechanisms of the adaptation.

## The slow adaptation of PNs correlates with a decrease in presynaptic vesicle release

The slow adaptation in PNs was also observed at low stimulus concentrations that activated only a single glomerulus, suggesting that the mechanism could be intrinsic to the single glomeruli and not due to computation across glomeruli. In order to more directly test synaptic activity, we used a post-synaptically targeted calcium sensors (*d*Homer-GCaMP) and a red fluorescent sensor for synaptic vesicle release (Syp-pHTomato) (*Pech et al., 2015*). The dynamics of postsynaptic calcium in PNs resembled that measured with the cytosolic calcium reporters (*Figure 6A*). Interestingly, we observed that vesicle release from ORNs monitored using Syp-pHTomato showed a slow decrease equivalent to the dynamics of the postsynaptic calcium (*Figure 6B*). To properly characterize these dynamics, we fitted a LN model to the steady-state Syp-pHTomato signal (*Figure 6D*) that accurately predicted the synaptic activity of glomerulus DM1 (*Figure 6E*). Because the Syp-pHTomato sensor

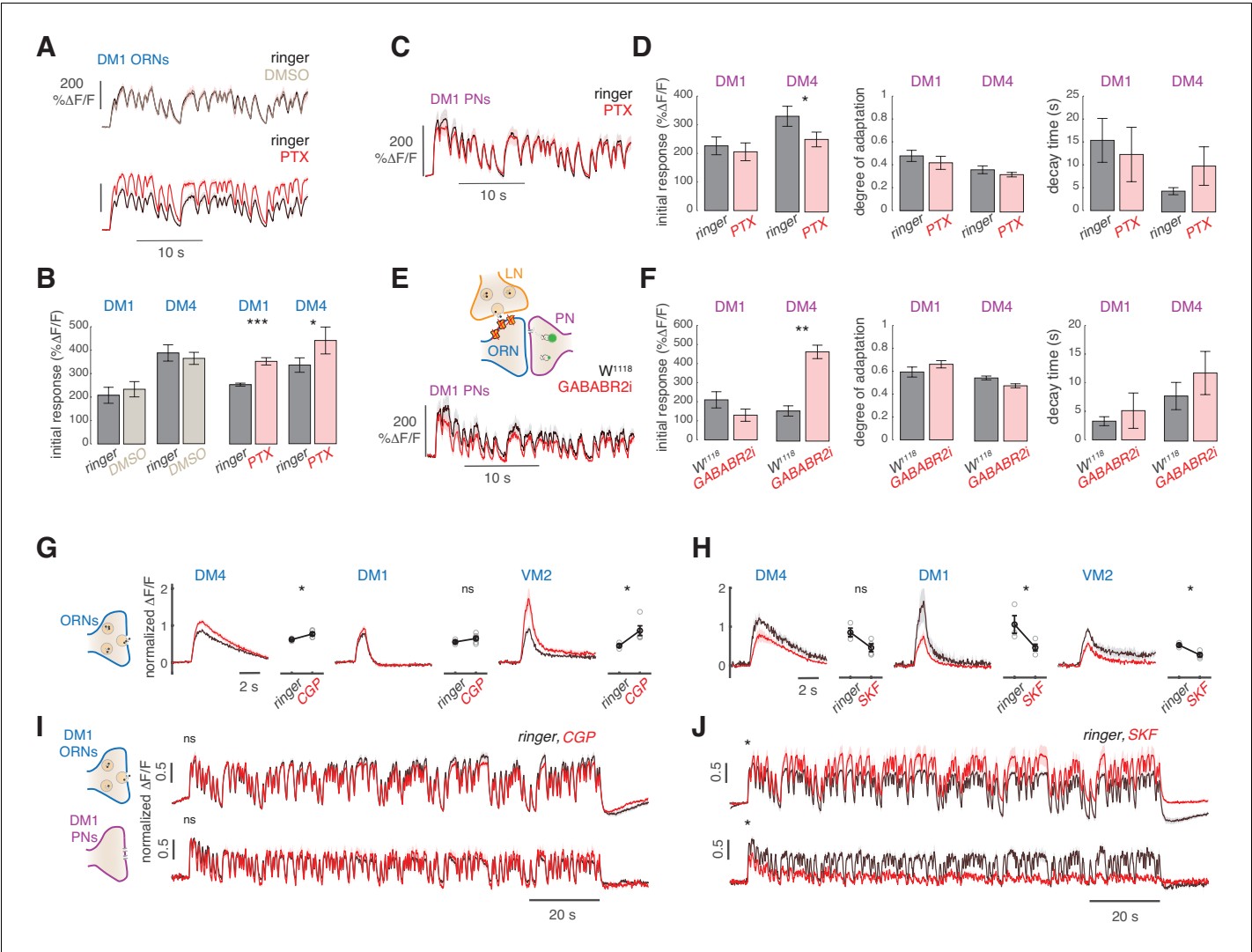

**Figure 5.** Slow adaptation in the PN calcium signal is not driven by lateral inhibition. (A–D) Application of the GABA-A receptor antagonist PTX does not affect calcium adaptation in PNs. (A) Application of 5 µM PTX enhances the calcium response from ORN terminals in glomerulus DM1. *Top:* Calcium response to randomly fluctuating odor stimulation (methyl acetate) before and after application of an equivalent DMSO control (n = 4). *Bottom:* Calcium response before and after application of 5 µM PTX (n = 6). Shaded areas indicate SEM. (B) PTX enhances the response to the first odor pulse measured from ORNs in glomeruli DM1 and DM4. Paired *t* test, *p<0.05, ***p<0.001. Error bars indicate SEM. (C) Mean calcium signals from PNs in DM1 in response to random stimulation before (black) and after (red) PTX application. Shaded areas indicate SEM (n = 10). (D) Initial response $r_0$, degree of adaptation $d$, and decay timescale $\tau$ of PN activity under sustained random stimulation for glomeruli DM1 and DM4 before and after PTX application, calculated by fitting an exponential decay function to the peak response to each odor pulse in the random sequence (*Figure 5—figure supplement 1*). Error bars indicate SEM. PTX application slightly decreases DM4 response but has no effect on the other parameters (paired *t* test, n = 10). (E) Schematic illustration of GABA-B-R2 RNAi-mediated down-regulation and simultaneous expression of the calcium reporter GCaMP3 in PNs. *Bottom:* Mean calcium signal from DM1 PNs in flies expressing GABA-B-R2-RNAi and in control w[1118] flies. Shaded areas indicate SEM (n = 5). (F) GABA-B-R2-RNAi expression affects the initial response $r_0$ but not the degree of adaptation $d$ and the decay timescale $\tau$ of PNs activity in glomeruli DM1 and DM4 (**p<0.01, Mann-Whitney U-test, n = 5). Also see *Figure 5—figure supplement 1*. (G) Mean response of ORNs from DM4 and DM1 (methyl acetate 10[-4.3] dilution) and VM2 (ethyl butyrate 10[-7.5]) to a 1 s pulse, in control flies (black) or in flies treated with 25µM CGP54626 (GABA-B antagonist). (*p<0.05, ***p<0.001, Kruskal-Wallis test, n = 3-5). In both test and control flies, the odor response was first quantified in normal ringer. The ringer was then replaced with new ringer in control flies and with ringer + CGP54626 in test flies. Odor response was tested again 8 minutes after drug application. The response in presence of the drug was then normalized to the response to the first presentation of the odor to account for fluctuation in the odor concentration. (H) Same as in (G) with 40µM SKF97541 (GABA-B agonist) (*p<0.05, **p<0.01, Kruskal-Wallis test, n = 3-4). (I) Mean response of ORNs and PNs from DM1 to a random stimulus sequence (methyl acetate 10[-4.3] dilution) in ringer (black) or in CGP54626 (red). Response is normalized in each fly to the mean response to the first odor pulse obtained in ringer. No change was observed after drug application (n=6-8), similarly to control flies treated with ringer (not shown). (J) Mean response of ORNs and PNs from DM1 to a random stimulus sequence (methyl acetate 10[-4.3] dilution)

*Figure 5 continued on next page*

*Figure 5 continued*

using ringer (black) or SKF97541 (red). Response is normalized in each fly to the mean response to the first odor pulse obtained in ringer. Response to the first odor pulse is lower in both ORNs and PNs after drug application (paired t-test, n=5-6).

DOI: https://doi.org/10.7554/eLife.43735.010

The following figure supplement is available for figure 5:

**Figure supplement 1.** Single trials from GABA experiments.

DOI: https://doi.org/10.7554/eLife.43735.011

had weaker baseline fluorescence intensity than the GCaMP-based calcium reporters, fewer glomeruli gave measurable responses. We monitored unambiguous and robust responses from five glomeruli (*Figure 6—figure supplements 1–2*). Then, we used the Bayesian Information Criterion (BIC) to select which of three models (a single exponential, an exponential plus a constant, or a double exponential function) best fits each linear filter (see Materials and methods). In all cases, the linear filter of the Syp-pHTomato signal could be fitted by a double exponential with a positive ($A_1 > 0$) and negative ($A_2 < 0$) component (BIC <0, *Figure 6D and F* and *Figure 6—figure supplement 1*). The positive component had a relatively fast timescale ($\tau_1 \sim 300–400$ ms, *Figure 6G*), similar to the calcium dynamics reported by GCaMP3 (*Figure 2E*). The negative component instead had a much slower timescale ($\tau_2$) of the order of ~10 s (*Figure 6G*). We reasoned that this slow negative component could be responsible for the slow adaptation observed in PNs (*Figure 4*). If this was the case, the slow timescale of the Syp-pHTomato should be glomerulus and concentration dependent, similar to what we observed for the calcium dynamics (*Figure 2E*). Indeed, it correlates significantly with the corresponding timescales fitted to the decay in cytosolic calcium in PNs (*Figure 6H*). This result support a model where slow depression of vesicle release from ORN terminals drives the slow adaptation observed in odorant-evoked calcium responses at PN postsynapses.

## Steady state calcium dynamics in ORNs and PNs

Next, we set out to quantify the steady state calcium dynamics in different glomeruli. ORNs calcium responses from different glomeruli were all sustained and could be fitted by monophasic linear filters both with GCaMP3 (*Figure 7A*) and GCaMP6f (*Figure 1—figure supplement 2A*). Some glomeruli had faster responses than others (e.g. DC1 and DL5). These differences were robust to deconvolution of the sensor kinetics (*Figure 7B*), demonstrating that the calcium dynamics reported by GCaMPs can be used for internal comparisons (between glomeruli or cell types). Deconvolved filters are subject to high-frequency noise; therefore, we used the data obtained with GCaMP3 for comparing glomeruli and stimuli. All ORNs filters could be fitted by a single exponential function (*Figure 7A*), apart from a constant which captured non-stationary features of nearly saturated responses (DL5 and DM4, *Figure 7A*). The fitted timescales ranged between 300 ms and 1 s, and we did not observe any consistent dependence of these timescales on response amplitude within single glomeruli (*Figure 7C,D*). A two-way analysis of variance suggested that the response timescale depended more strongly on glomerulus type ($p < 10^{-6}$) than on concentration. The timescales inferred from calcium responses should be considered as upper bounds of the real timescale since the measured calcium signals are limited by the kinetics of the sensor. We found similar results (glomerulus specific and concentration independent timescales) repeating these experiments with GCaMP6f (*Figure 1—figure supplement 2B*).

*Drosophila* PNs respond faster than ORNs to odor stimuli (*Bhandawat et al., 2007*) as a result of short-term synaptic depression (*Kazama and Wilson, 2008*). As previously shown in second-order olfactory neurons firing activity (*Geffen et al., 2009*; *Gupta et al., 2015*; *Jacob et al., 2017*), after the initial adaptation period, odor-elicited calcium responses could also be described by an LN model (*Figure 7E*). The linear filter of the PNs is not monophasic, as for the corresponding ORNs, but shows a negative component (see DM2 in *Figure 7A and E*). This biphasic filter is consistent with short-term depression at the ORN-PN synapses (*Kazama and Wilson, 2008*) and indicates that PNs further differentiate incoming odor signals (*Cafaro, 2016*; *Kim et al., 2015*). Importantly, the PNs linear filters gradually shifted from biphasic to monophasic as the concentration increased (e.g. DM1 in *Figure 7E*). These concentration dependent differences were observed also after deconvolution of the sensor kinetics (*Figure 7F*).

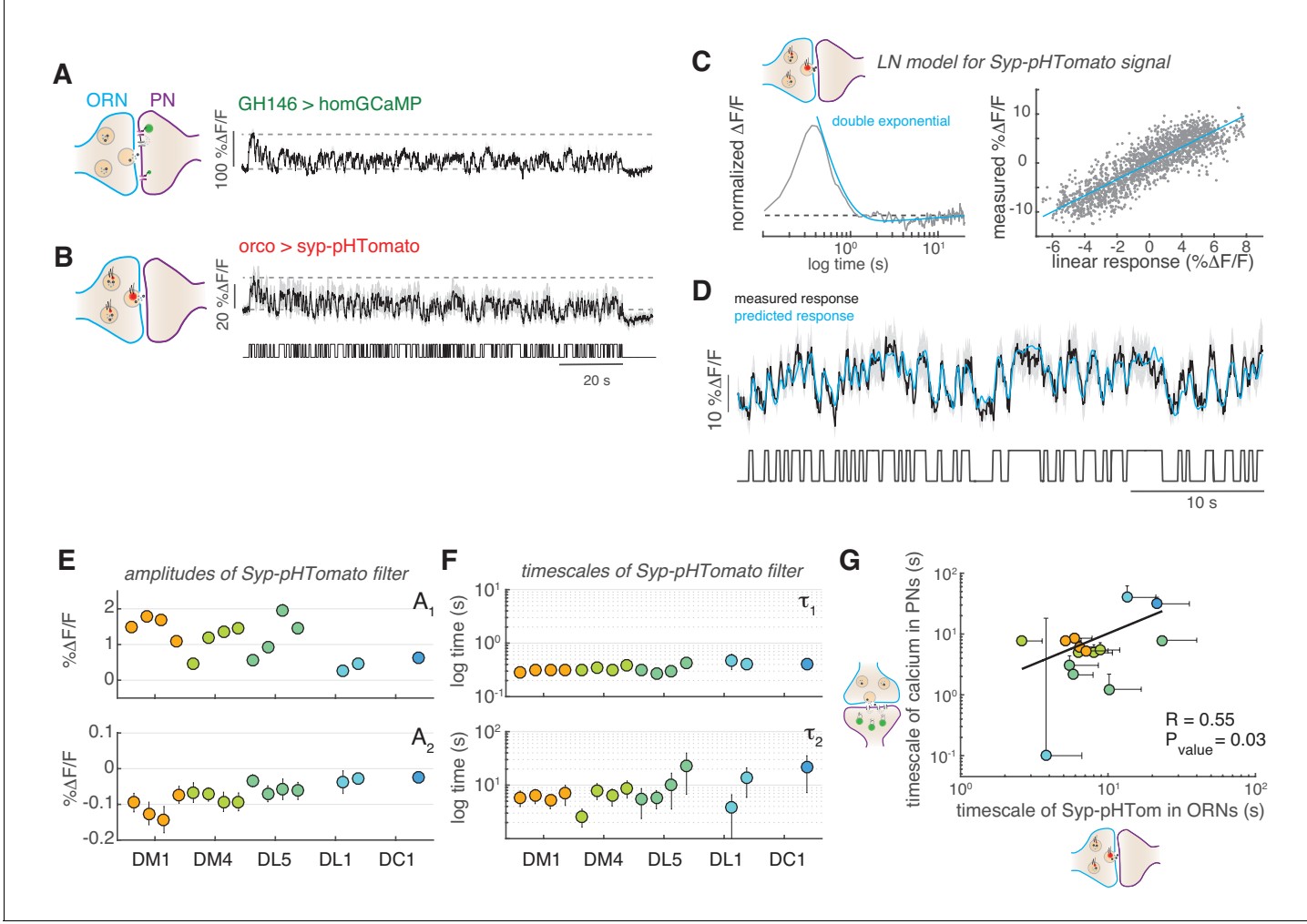

**Figure 6.** Slow adaptation in the PN calcium signal reflects slow depression in vesicle release at the ORN terminals. (A) Postsynaptic calcium signal in PNs measured with dHomer-GCaMP from glomerulus DM1 in response to a fluctuating odor stimulus. The gray-shaded area represents SEM. (B) Fluorescence signal of Syp-pHTomato from ORNs in DM1. (C) LN model for the Syp-pHTomato signal. *Left:* Normalized linear filter fitted to the data in (E) (gray) and exponential fit to the linear filter (cyan, double exponential). *Right:* Scatter plot of the predicted linear response and measured *response* (gray) and corresponding linear fit (cyan). (D) Mean Syp-pHTomato fluorescence (black, as in (C)) and response predicted by the LN model (cyan). (E) An LN model was fitted to all responses measured in five glomeruli (color coded as before) at four stimulus concentrations. The BIC was used to choose whether the linear filter shape was better captured by a simple exponential or a double exponential function. In all cases, a double exponential was selected $k_3(t) = A_1 e^{-\frac{t}{\tau_1}} + A_2 e^{-\frac{t}{\tau_2}}$ with $A_1 > 0$ and $A_2 < 0$ (see also *Figure 6—figure supplement 1*). The response of DL1 to $C_{01}$ and $C_{08}$ and the response of DC1 to $C_{01}$, $C_{08}$, and $C_{13}$ were too weak to fit a model. (F) The timescales of the negative exponential $\tau_2$ are always slower than that of the positive exponential $\tau_1$, indicating a slow adaptive process. (G) The slow timescale $\tau_2$ of the Syp-pHTomato signal correlates with the timescale of the slow adaptation measured in the calcium signal of postsynaptic PNs (*Figure 4G*). Different colors indicate different glomeruli. Error bars indicate 95% confidence intervals.

DOI: https://doi.org/10.7554/eLife.43735.012

The following figure supplements are available for figure 6:

**Figure supplement 1.** Linear filters for Syp-pHTomato signal from ORN terminals.

DOI: https://doi.org/10.7554/eLife.43735.013

**Figure supplement 2.** Linear correction to Syp-pHTomato fluorescence.

DOI: https://doi.org/10.7554/eLife.43735.014

The LN model captured most of the variance in the data equally well at low and high concentrations (*Figure 7—figure supplement 1*), although it failed to capture a drop in the calcium signal induced by longer pulses (*Figure 7—figure supplement 2A–C*). Again, we used BIC to select the most parsimonious model that fits the linear filters. For several glomeruli, the best-fitted function

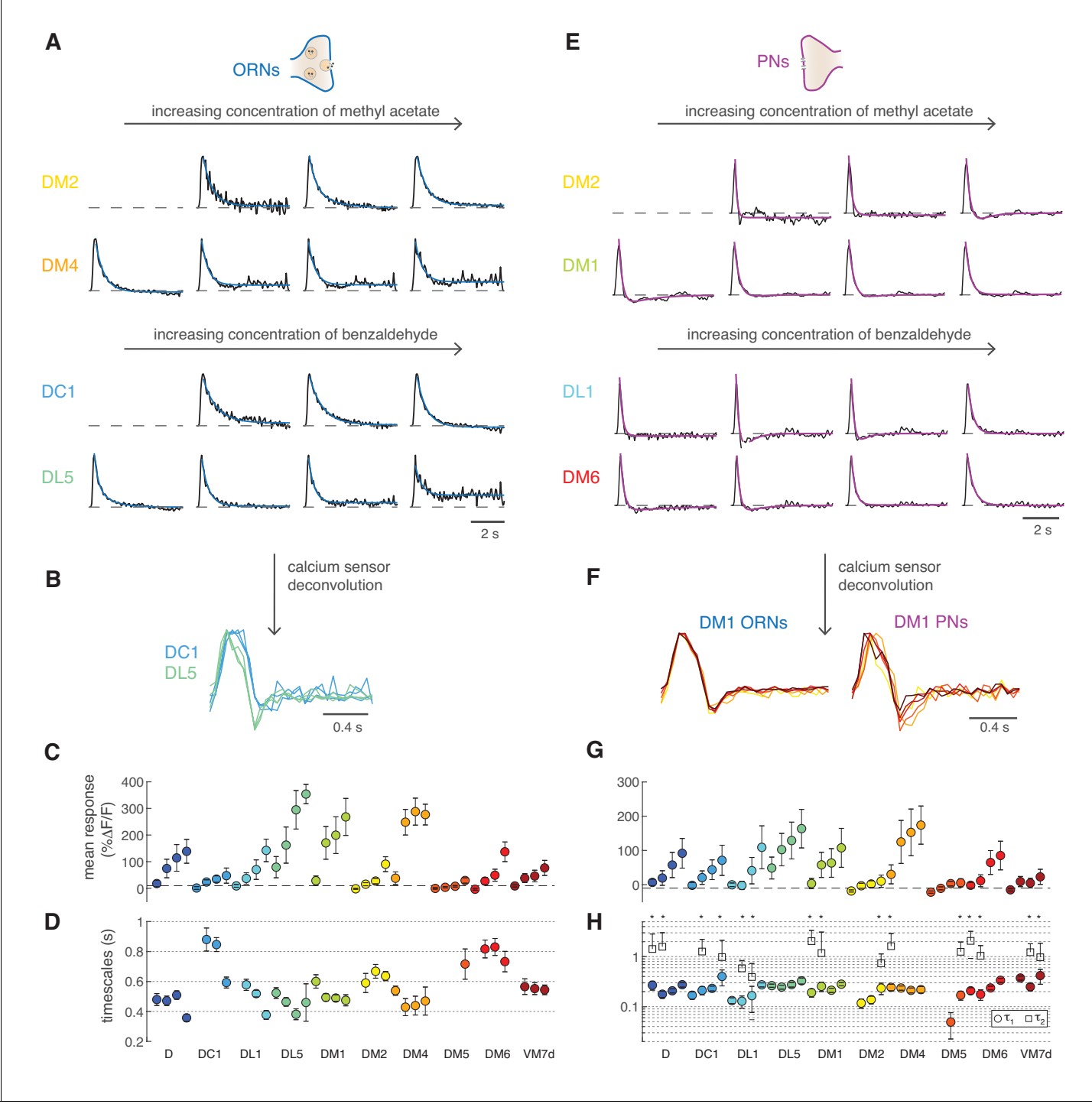

**Figure 7.** Calcium dynamics in ORNs and PNs. (**A**) Linear filters (black) and corresponding exponential fit (cyan) for ORNs in four glomeruli at four concentrations of the stimulus. Two odors were used (methyl acetate and benzaldehyde) that capture the response of two non-overlapping sets of glomeruli. Responses below 10% ΔF/F were removed from further analysis because of an insufficient signal-to-noise ratio (n = 4–11). (**B**) Example of the linear filters obtained after deconvolution of the calcium sensor, color indicates glomerulus identity and different curves correspond to different stimulus intensities. (**C**) Mean responses to the fluctuating stimulus averaged over time for 10 glomeruli (color coded) at four stimulus intensities (same color). Error bars indicate standard deviations. Dashed lines indicate a threshold of 10% ΔF/F. (**D**) Timescale $\tau_1$ of the linear filter resulting from the exponential fit. Error bars indicate 95% confidence intervals. Timescale of the linear filters is significantly anticorrelated to mean response ($R = -0.6$; p<0.0004), glomerulus dependent (p<$10^{-6}$) and slightly dependent on stimulus intensity (p<0.02; two-way analysis of variance). Sample size per glomerulus was: $n_D = 6$, $n_{DC1} = 7$–8, $n_{DL1} = 7$–8, $n_{DL5} = 7$–8, $n_{DM1} = 9$–11, $n_{DM2} = 9$–11, $n_{DM4} = 9$–11, $n_{DM5} = 7$–8, $n_{DM6} = 7$–8, and $n_{VM7d}$=4–6. (**E**) Linear

*Figure 7 continued on next page*

*Figure 7 continued*

filters (black) and corresponding exponential fit (purple) for PNs in four glomeruli at four concentrations of the stimulus. For each glomerulus, three models were fitted (single exponential, single exponential plus a constant, and double exponential, see Materials and methods). The model with the lowest BIC was selected. The shape of the linear filter in PNs depends on stimulus intensity. (F) Linear filter obtained from calcium responses after deconvolution of the sensor kinetics, showing that PNs filters are concentration dependent. Color indicates concentration as in *Figure 2D*. (G) Same as in C for PNs. (H) Timescales of the linear filter resulting from the exponential fit. Stars (*) indicate data sets that were fitted by a double exponential (model 3): in this case, a second timescales $\tau_2$ is shown, which is associated to the negative lobe pf the double exponential. $\tau_1$ is slightly correlated with the mean response ($R = 0.4$, p<0.02), whereas $\tau_2$ is independent from it (p>0.05). Differences between glomeruli are not significant (Kruskal-Wallis test, p>0.05). The sample size per glomerulus was as follows: $n_D = 7$, $n_{DC1} = 3–4$, $n_{DL1} = 6–7$, $n_{DL5} = 7–8$, $n_{DM1} = 6–7$, $n_{DM2} = 6–7$, $n_{DM4} = 6–7$, $n_{DM5} = 5–7$, $n_{DM6} = 7–8$, and $n_{VM7d} = 6–7$.

DOI: https://doi.org/10.7554/eLife.43735.015

The following figure supplements are available for figure 7:

**Figure supplement 1.** Residual-to-noise ratio in LN model cross-validation.

DOI: https://doi.org/10.7554/eLife.43735.016

**Figure supplement 2.** LN model for PN response.

DOI: https://doi.org/10.7554/eLife.43735.017

was a double exponential with a slow negative component (*Figure 7G,H*, $\tau_2$ ~1.2 s). However, we found a significant dependence of the model selected on the response amplitude: the filter shape was usually monophasic at very low responses, biphasic at intermediate responses, and then again monophasic at high responses (*Figure 7—figure supplement 2D*, Kruskal-Wallis test, p<0.02). A diversity of filter shapes has been reported in locusts, moths, and rats (*Geffen et al., 2009*; *Gupta et al., 2015*; *Jacob et al., 2017*), although responses in these systems could not be assigned to specific glomeruli. One possible interpretation of our results is that at low responses there is little synaptic depression between ORNs and PNs, which becomes stronger at intermediate stimulus intensity, making the PN response more biphasic and differentiating. Stronger responses probably engage additional mechanisms (lateral inputs or postsynaptic modulation) with their own dynamics (*Geffen et al., 2009*) and also drive the response towards saturation. Summing feedforward ORN inputs through depressing synapses with stimulus-specific lateral inputs certainly induce strong non-linearities in the PN output dynamics across concentrations (*Figure 7—figure supplement 2A–C*), similarly to what found in mitral/tufted cells (*Gupta et al., 2015*). In conclusion, the LN model confirms that PNs calcium responses are more adaptive than ORNs, and the specific dynamics (filter shape) of different glomeruli are concentration dependent.

## Slow adaptation linearly rescales the combinatorial odor representation in PNs

We then asked what the slow adaptation of PN activity implies for the combinatorial odor representation in the population of PNs. For each glomerulus and concentration, we quantified the peak response to each of the odor pulses in the random sequence (as in *Figure 4*) and used it to compare the combinatorial odor representations in ORNs and PNs using principal component analysis (PCA). Combinatorial ORN responses were stationary over a time period of 2 min and, consequently, the population responses to the first odor pulse were very similar to the responses to a pulse 2 min later (*Figure 8A*). On the contrary, the combinatorial representation of the stimulus changed over time in PNs (*Figure 8B*). Since the calcium signal from single glomeruli decreased proportionally to the peak response (*Figure 4*), the combinatorial activity represented by the PC1 scaled linearly and resulted in a multiplicative rescaling of the ORN-to-PN transformation (*Figure 8C*). This principle of scaling was confirmed using a second odorant, benzaldehyde (*Figure 8D*). These results show that slow synaptic adaptation adjusts the PN response, while preserving the odor representation at the population level.

## Combinatorial representations of fluctuating odor stimuli from ORNs to PNs at steady state

Last, we asked how the different response properties of ORNs and PNs affected the steady state combinatorial representation of the odorant at the population level. Increasing the concentration of an odorant led to more active glomeruli by recruiting ORNs that express receptors with lower

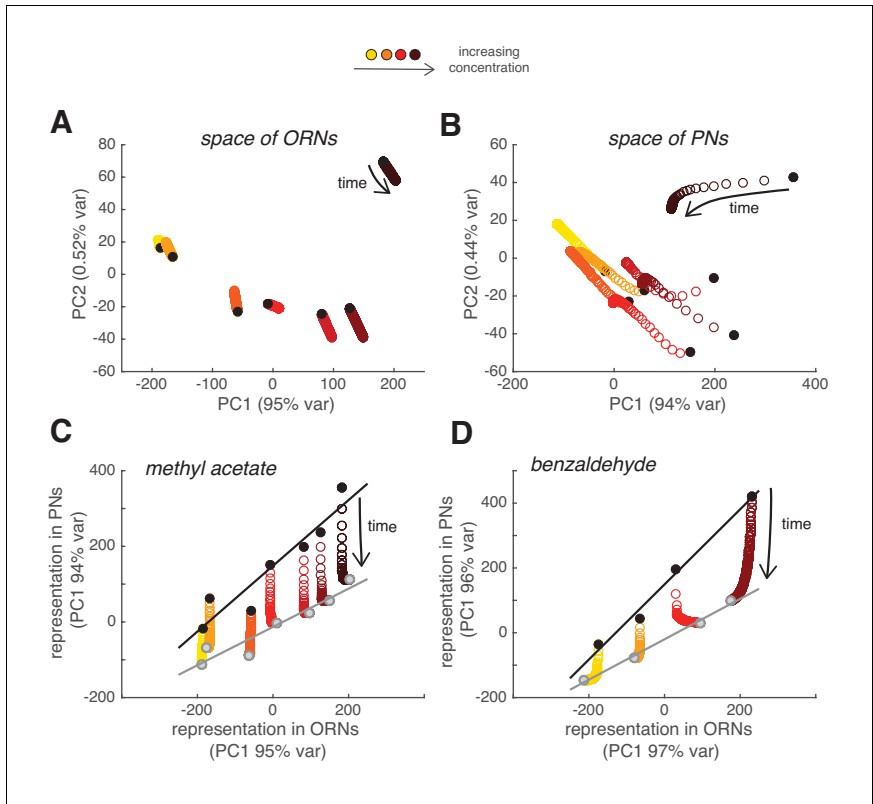

**Figure 8.** Slow depression linearly rescales odor representations in PN space. (**A**) Combinatorial representation of methyl acetate represented by the first and second principal components of the ORN activity. Black circles correspond to the response to the first pulse in the random sequence; all other partially overlapping empty circles correspond to the response to subsequent odor pulses, showing how ORN odor representation is stable over 2 min. (**B**) Same as (**A**) for the response of PNs, showing how odor representations in PN space change over time. (**C**) Transformation of spatial representation from ORNs to PNs. Each circle represents the combinatorial response for each single odor pulse of PNs versus ORNs quantified by the first principal component. Black/gray circle: response to first/last odor pulse. Black and gray lines are linear fits to the initial and final response. (**D**) Same as in (**C**) but for benzaldehyde.

DOI: https://doi.org/10.7554/eLife.43735.018

The following figure supplement is available for figure 8:

**Figure supplement 1.** Contribution of different glomeruli to PCA Relative contributions of the different glomeruli to the PC1, here caclulated using the peak response to the single pulses as in *Figure 8*.

DOI: https://doi.org/10.7554/eLife.43735.019

odorant affinities (*Figure 9A*). As the concentration increased, the response of ORNs in individual glomeruli saturated and their dynamic range shrank, as shown by the large offset in the static nonlinearity for high concentrations (*Figure 9B*, top). However, responses of the corresponding PNs spanned a wider dynamic range (*Figure 9B*, bottom). This suggests that, in ORNs, the coding of stimulus fluctuations might shift from one subset of glomeruli to another as the odorant concentration changes while being more distributed in the population of PNs (*Figure 8—figure supplement 1*). As shown above (*Figure 8*), the representation of a fluctuating stimulus was mainly unidimensional in the glomerular space. For increasing odorant concentrations in ORNs, the dynamic range of the combinatorial representation shifted along this single dimension (x-axis, *Figure 9C*), whereas in PNs the dynamic range stretched (y-axis, *Figure 9C*). Accordingly, the variance of the response scaled differently between PNs and ORNs (bars, *Figure 9C*). In our experimental design, the standard deviation ($\sigma$) and the mean ($m$) of the random stimulus scaled linearly with the odorant concentration. As a consequence, the coefficient of variation of the stimulus ($\mathrm{CV} = \frac{\sigma}{m}$) was constant across concentrations ($\mathrm{CV} = 0.9$). The CV of the neural response is expected to be lower because of the

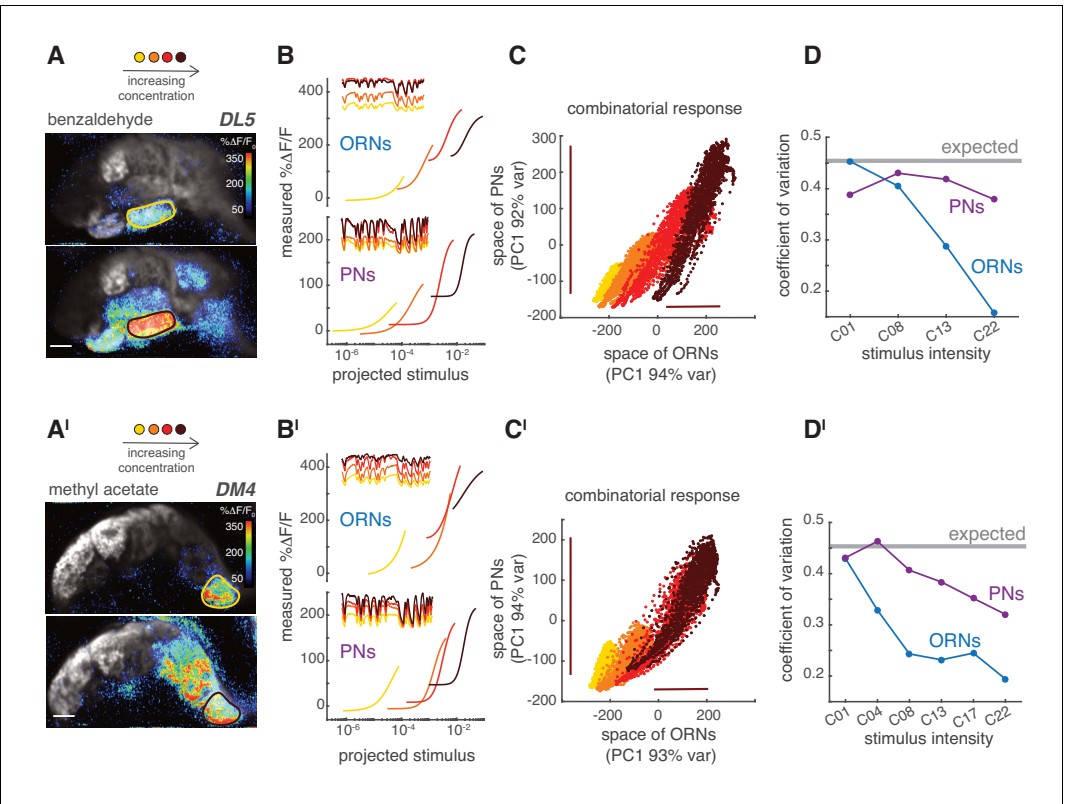

**Figure 9.** PNs accurately encode stimulus variance. (**A**) Calcium increase in ORN terminals upon stimulation with benzaldehyde at two different concentrations ($C_{01}$, $C_{22}$). (**B**) Static non-linear functions fitting the relationship between measured and predicted linear calcium responses for ORNs (top) and PNs (bottom) in glomerulus DL5. As odorant concentration increases (yellow → dark red), the responses of both ORNs and PNs become more non-linear and saturates. However, PNs maintain a broader dynamic range compared to the corresponding ORNs, even when these saturate (dark red). Note that here the same binary stimulus was used for all concentrations to visualize the change in dynamic rage. (**C**) Scatter plot of the first principal components of the combinatorial odor representation of the odor stimuli in ORNs and PNs. Each dot represents the measured response at each time point of a 2-min-long pseudorandom stimulus. The combinatorial odor representation in the space of five responding glomeruli was almost unidimensional in both ORNs and PNs with the first principal components explaining >90% of the variance. Colors indicate odorant concentration (yellow → dark red). Bars indicate the variation in ORN and PN responses to the stronger stimulus intensity as a visual estimate of the response variance. (**D**) CV as a function of stimulus intensity, quantified as the standard deviation divided by the mean of the first principal component of the population response. The gray line indicates the expected CV (see Materials and methods). (A'–D') Same as (A–D) but for methyl acetate as odorant.
DOI: https://doi.org/10.7554/eLife.43735.020

neuron integration time (CV = 0.45; see Materials and methods), but still independent of concentration if the coding of stimulus fluctuations was accurate. Interestingly, we found that although the CV dropped from 0.45 to 0.15 in ORNs as concentration increased, it stayed fairly constant in PNs (*Figure 9D*). These results were valid for a second odorant (*Figure 9A'-9D'*). We conclude that adaption in the responses of single PN types allows the glomerular PN populations to robustly encode both the mean and the variance of the stimulus over a larger range of concentrations than ORNs.

## Discussion

### Firing versus calcium adaptation in ORNs

Along a sensory pathway, multiple mechanisms can contribute to adaptation. However, identifying the contribution of individual processes is challenging. Here, we used imaging of synaptic activity to

isolate the mechanisms underlying adaptation along the olfactory pathway in *Drosophila*. Our experiments revealed that presynaptic calcium signals from the primary olfactory sensors, the ORNs, as reported by fluorescent indicators, do not show the same adaptation properties as the firing measured in the dendrites of the same neurons. How to explain these findings?

On short timescales (~100 ms), the true calcium dynamics are certainly affected by the sensor kinetics and a linear deconvolution (*Figure 2*) recapitulates some of the phasic properties of the firing rate. This kind of approach is reasonable, but not exhaustive as it doesn't take into account possible non-linearities in the sensor kinetics. However, a direct comparison between firing rate and calcium response to isolated odor pulses (*Figure 3*) demonstrates a striking correspondence in two neuron types, suggesting a similar working range of the sensor in these two cell types. This is in general an assumption when comparing calcium dynamics across neurons (*Si et al., 2019*). As it is difficult to make a cell-specific calibration of the sensor in vivo, the same model is assumed for all conditions and therefore deconvolution leaves cell- or stimulus-specific differences in kinetics unaffected (*Figures 2E,I* and *7B,F*).

Following the phasic response, adaptation should drive a change in response properties, usually a decreased responsiveness. Here, we have investigated changes in response gain using both a sustained flickering stimulus and a steady background stimulation. In both cases we do not observe the changes expected from firing rate activity. How could that be?

Inhibitory lateral inputs have their own transient dynamics (*Nagel et al., 2015*), therefore one possibility could be that inhibition suppresses the initial calcium response stronger than the later response. This later relief from inhibition could account for an overall sustained response. However, we observed sustained activity in ORNs even at very low concentrations that activate a single glomerulus and the sustained ORN response could not be altered by blocking GABA receptors.

Another possibility is that not all presynaptic calcium measured in our experiments is involved in vesicle release. For example, there could be two components, one that follows firing activity and allows the release of vesicles and a second one involved in vesicle recycling (*Rizzoli, 2014*). However, both should be driven by the odor stimulus and have perfectly balanced dynamics, one adaptive and one facilitating so that the summed calcium is tonic. The other possibility is that calcium at the ORNs presynapses is regulated in a way to extract certain features of the stimulus from firing activity, either by lateral inputs or cell autonomously, for example through expression of different voltage-gated calcium channels (*Gu et al., 2009*; *Iniguez et al., 2013*), activity-dependent modulation of their expression (*Gratz et al., 2018*) and function (*Dason et al., 2012*).

These two possibilities do not only differ at a mechanistic level, but also at the functional level. While it remains certainly true that the firing rate follows the Weber-Fechner law and adapts to stimulus mean and variance (*Gorur-Shandilya et al., 2017*), the implications for coding are different depending on whether these properties are transmitted downstream from the antenna or whether they play a different function in calcium regulation at the presynapses. Further mechanistic insight should be gathered in the future by investigating the underlying molecular mechanisms, for example calcium channels expressed in these neurons and possible modulators of their activity.

## Synaptic depression as a mechanism for adaptation

Previous studies have suggested that the PN adaptive capabilities rely mostly on properties of the ORN-PN synapses (*Cafaro, 2016*; *Kazama and Wilson, 2008*; *Nagel et al., 2015*). Here, we show that a combination of slow and fast dynamics occurs at these synapses that allows the population of PNs to maintain a neural representation of the stimulus identity and statistics (i.e. mean and variance). On short timescales, PNs calcium responses can be modeled by fast biphasic linear filters that underlie their capability to sense fast changes in concentration (*Bhandawat et al., 2007*; *Kim et al., 2015*). This biphasic response is consistent with the property and function of previously described short-term presynaptic depression, a mechanisms that enhances neural temporal coding (*Tsodyks and Markram, 1997*) as reported also in other insects (*Huston et al., 2015*).

Here, using an optical imaging approach, we identified a slower component in the dynamics of the presynapses that would otherwise be difficult to isolate. pH-dependent sensors of synaptic vesicle release have been previously used in both vertebrates (*Dreosti et al., 2009*; *Miesenböck et al., 1998*) and insects (*Ng et al., 2002*; *Pech et al., 2015*; *Yu et al., 2004*). In particular, they made it possible to characterize synaptic depression and facilitation that mediate adaptation in the visual system (*Nikolaev et al., 2013*). These forms of short-term plasticity result mostly from an interplay

between calcium-dependent modulation of the probability of vesicle release and the depletion of available vesicles. The simplest model of synaptic depression consists of a finite pool of ready-releasable vesicles that is replenished from an infinite pool at a certain limiting rate (*Abbott, 1997*). However, synaptic depression can be more complex than that. Synaptic depression at the neuromuscular junctions of *Drosophila* larva can be described by a model in which the supply pool is finite and is itself replenished at slower time-scales (~10 s) by a larger reservoir (*Hallermann et al., 2010*). However, other models are also possible and our current understanding of ORN-to-PN synapses is insufficient to point to a specific one.

The interplay between feedforward depression and lateral inputs on slow timescales remains to be further investigated. Although our analysis excludes a direct role of inhibition on the PNs adaptation, inhibitory presynaptic inputs do play a role in modulating depression (*Olsen and Wilson, 2008*). However, in presence of a sustained stimulus that activates multiple glomeruli, tonic activation of the GABA-ergic pathway does not simply decrease presynaptic activity releasing depression, as one would expect, rather it disrupts the entraining of the network with the stimulus, resulting in a loss of temporal resolution and response in PNs (*Figure 5J*). It is likely that under sustained stimulation, changes in the activity of LNs (*Nagel et al., 2015*) as well as recurrent connections within the AL could affect the coupling between inhibition and depression at the ORN presynapses. Finally, it remains unclear whether the slow adaptation described here can be solely explained by presynaptic mechanisms or whether involves post-synaptic dynamics as well (*Cafaro, 2016*; *Nagel et al., 2015*).

## Fast and slow adaptation across sensory modalities

In different sensory modalities across species, adaptation to stimulus statistics occurs not only at different levels along sensory pathways but also on different timescales. For example, retinal ganglion cells adapt their sensitivity to a change in the visual stimulus on both short (<100 ms) and slow timescales (~10 s) (*Chander and Chichilnisky, 2001*). Similarly, auditory neurons in songbirds slowly adapt their firing to stimulus variance, but instantaneously adapt their filter shape and gain (*Nagel and Doupe, 2006*). These two forms of adaptation play different roles in sensory coding. Fast adaptation acts as a gain control mechanism that prevents saturation by decreasing sensitivity, while slow adaptation allows adjustment to the overall variance of the stimulus (*Baccus and Meister, 2002*). Adaptation can, in principle, occur on a continuum of timescales, from milliseconds to minutes (*Fairhall et al., 2001*) and involve several mechanisms. It can rely on physiological mechanisms within a single cell (*Toib et al., 1998*) or, alternatively, result from neuronal network dynamics. For example, potassium conductance mediates slow contrast adaptation in visual cortical neurons by a purely cellular mechanism (*Sanchez-Vives et al., 2000*). Network mechanisms can explain slow adaptation in ganglion cells as a result of reduced presynaptic glutamate release (*Manookin and Demb, 2006*). Both depression and facilitation mediate adaptation to contrast in the zebrafish retina (*Nikolaev et al., 2013*). Here, we report that not only fast, but also slow adaptation in the olfactory PNs of *Drosophila* can be attributed to depression of presynaptic activity. Our data, together with previously reported fast and slow postsynaptic currents (*Nagel et al., 2015*), support a major role of the synapse in adaptive coding. These findings do not exclude that additional network or cellular mechanisms are involved in adaptation on different timescales or under different stimulus conditions.

## Divisive normalization and variance coding

Previous studies have identified divisive normalization as a neural computation implemented in the antennal lobe by a network of inhibitory local interneurons (*Olsen et al., 2010*), which scales the response of single neurons by the overall population activity (*Carandini and Heeger, 2011*). It can serve a diversity of functions, but in the olfactory system it is believed to enhance the dynamic range of PNs (*Galizia, 2014*; *Wang, 2012*). How do our results relate to these findings? *Root et al. (2008)* have shown that GABA and the expression of GABA-B receptors in ORNs mediate this lateral inhibition by modulating calcium responses in ORN terminals. Therefore, presynaptic lateral inhibition should not be the main determinant of the ORN-to-PN transformation described here, although it of course affects presynaptic activity. This is consistent with our finding that manipulations of GABAergic pathways do not interfere with the slow adaptation of PNs. We conclude that divisive normalization and slow depression are two complementary mechanisms that adjust the PNs dynamic range.

While the lateral inhibition scales responses based on overall network activity (*Olsen et al., 2010*), the slow depression described here adjusts the response gain to the statistics (mean and variance) of the incoming stimuli. Moreover, lateral inhibition depends on the connectivity of the AL network and, therefore, affects different glomeruli differently. On the contrary, the slow adaptation depends on feed-forward connectivity and proportionally scales combinatorial odor presentations. It remains unclear whether lateral inhibition contribute to this proportional scaling of the response through a stimulus-driven change in the LNs response (*Nagel et al., 2015*). Finally, lateral inhibition might also play an important role in the context of cross-adaptation between different odors and deserve further attention.

### Consequences for population coding and behavior

The perceptual effects of adaptation as a decreased sensitivity to the stimulus are usually more evident in presence of a sharp change in the stimulus properties. A recent work on odor driven walking behavior has shown that flies respond with an increase in upwind velocity to odor onset which adapts on a timescale of ~10 s (*Álvarez-Salvado et al., 2018*). This adaptation timescale cannot be explained with ORN firing rate adaptation (that is much faster ~100 ms), but it is surprisingly in agreement with the slow presynaptic depression identified here. Animals that use olfaction to localize a source will likely implement different strategies depending on the statistics of the odor landscape (*Baker et al., 2018*; *Gaudry et al., 2012*) and the availability of different timescales for the sensory processing of the stimulus may allow them to flexibly adapt their search strategy.

On the other hand the identity of a stimulus must be maintained as neural activity when the stimulus statistics change or the sensory system adapts to sustained stimuli (*Dean et al., 2005*; *Gutnisky and Dragoi, 2008*). Here, we have demonstrated that a linear combination of glomeruli activity represents the stimulus identity during a highly dynamic stimulation and this combination scales proportionally for different mean values of this stimulus. This combinatorial representation is stationary over time in ORNs, but undergoes a slow rearrangement in the space of PNs. This rearrangement consists of a linear scaling of population activity that preserves information about both stimulus identity (same population of neurons) and intensity (same order of stimuli). Animals that navigate a turbulent odor landscape will experience increased stimulus variance and increased stimulus intensity as they approach the odor source (*Celani et al., 2014*). The mechanisms described here can explain how odor perception is preserved while the animal moves toward an odor source and the sensory system dynamically adjusts its activity to better represent the statistical context of the odor signal.

## Materials and methods

### Fly strains

Flies were raised on cornmeal food at 25°C and 60% humidity under controlled 12 hr light/dark cycles. All experiments were performed at room temperature (20°C) on female flies that were 6 to 7 days old. Fly strains used were as follows: *UAS-GCaMP3* (*Tian et al., 2009*), *UAS-GCaMP6f* (*Chen et al., 2013*), *or83b-GAL4* (*Larsson et al., 2004*), *GH146-GAL4* (*Stocker et al., 1997*); *UAS-Syp-GCaMP*, *UAS-Syp-pHTomato*, *UAS-dHomer-GCaMP*; *+/Or67a-mCherry* (*Pech et al., 2015*), *GABABR-R2 RNAi/CyO*; *GABABR-R2 RNAi/TM6B* (*Root et al., 2008*), *GH146-LexA/Gla; LexAop-GCaMP3/TM6* (*Martelli et al., 2017*).

### Calcium imaging

The calcium fluorescent reporter GCaMP3 was used throughout the paper (unless otherwise specified) to allow comparison with synaptically tagged sensors (*Pech et al., 2015*). Flies were briefly anesthetized on ice and fixed on a custom-built holder that left the antennae exposed to air. The head capsule was opened and covered with *Drosophila* ringer (5 mM Hepes, 130 mM NaCl, 5 mM KCl, 2 mM MgCl2, 2 mM CaCl2, 36 mM sucrose, pH 7.3). For optical imaging, we used a LSM7 two-photon microscope (Zeiss) equipped with a mode-locked Ti-sapphire Chameleon Vision II laser (Coherent), with a Plan-Apochromat 20x/1 water-immersion objective (Zeiss) and a set of 500 to 550 nm and 600 to 680 nm bp filters and a 600 nm dichroic mirror. The excitation wavelength was 920 nm and images were recorded at 20 Hz with a resolution of 0.21 µm/pixel.

In pilot experiment (whose data are not included in the manuscript), we determined that a sample size of n = 5 gives between 10% and 20% variation in the measured fluorescence signals in response to an odor (standard deviation relative to mean activity). Therefore, we decided to have a minimum n = 5 across all experiments. For those experiments that required the identification of multiple glomeruli in the same animal, it was necessary to acquired data from about 9–11 animals: the final sample size varies depending on whether specific glomeruli were identifiable in the different preparations. The detailed information about sample size can be found in the figure legend. Glomerulus identification was based on *Grabe et al. (2015)* and *Münch and Galizia (2016)*. In all the experiments, the same stimulus was presented only one to the same animal in the same condition, therefore we did not average on technical replicates. All the attempts to reproduce the results in our laboratory (biological replicates) were successful. Experiments in *Figures 2B*, *3* and *5G–L* and in *Figure 1—figure supplement 2* and *Figure 3—figure supplement 1* were performed at the University of Konstanz using a LSM5 two-photon microscope (Zeiss) with similar settings. Experiments performed on both setups yielded similar results (see *Figure 7A* and *Figure 1B*).

Flies were excluded from analysis only when the excessive movement of the in vivo preparation (muscle contraction or drifting of the focal plane) did not allow evaluation of the response for the full duration of the stimulus.

Odor concentrations were presented in a random sequence, and the random sequence was the same across different animals. When comparing the response of flies of different genotypes, measurements were always carried in parallel by alternating the genotype of tested flies until the desired sample size was reached.

## Electrophysiology

Single sensillum recordings were performed as previously described (*de Bruyne et al., 2001*; *Martelli et al., 2013*) using a silver-chloride electrode and glass pipettes filled with sensillum lymph ringer. Electrical signals were amplified (X1000) using an extracellular amplifier (DPA-2FL from npi Electronics) coupled with SH16-IZ head stage (Tucer Davis Technologies), bandpass filtered (300–5000 Hz), digitized at 25 KHz (CED) and acquired in Spike2 software. Spikes were sorted using a custom MATLAB routine. The same odor delivery system was used for both imaging and electrophysiology.

## Odor delivery

Flies were exposed to a continuous clean air stream (1 L/min), in which an odorous stream (0.1 L/min) was redirected through a solenoid valve (LEE) placed 5 cm from the exit of the delivery tube. The odor stream was kept at equilibrium with the liquid phase and continuously removed through a large suction tube to keep the experimental chamber clean. We used four mass flow controllers (MFCs; Alicat Scientific) to create gas dilutions of the odorant. Solenoid valves and MFCs were remotely controlled in MATLAB (Mathworks) through an Arduino board. 20 mL of odorant were prepared in high concentration ($10^{-2}$ volumetric ratio in paraffin oil) in 150 mL glass bottles placed on a magnetic stirrer. Odor stream concentration was left to equilibrate for 20 min at the minimum flow ($10^{-4}$ mL/min). The ratio of the flow through the four MFCs was adjusted remotely to obtain the desired concentration. Once a ratio was set, the odor flow was left to equilibrate for 3 min. Then, a 2-min-long pseudorandom stimulus was applied. The stimulus was generated by randomly choosing the state of the valve (open/closed) every 300 ms. Different odor concentrations were presented to the same animal in random order with a 3 min interval between them. Final gas volumetric dilutions used in the experiment were as follows: $C_1 = 10^{-5}$, $C_2 = 10^{-4.3}$, $C_4 = 10^{-4}$, $C_5 = 10^{-3.7}$, $C_8 = 10^{-3.3}$, $C_{13} = 10^{-3}$, and $C_{22} = 10^{-2.3}$. The reproducibility of the odor stimulation was confirmed using a photoionization detector (Aurora) (*Figure 1—figure supplement 1*). For the experiments in 2B, 3, 5 G-L and in *Figure 1—figure supplement 2* and *Figure 3—figure supplement 1a* similar delivery system was developed at the University of Konstanz using Analyt-MTC mass flow controllers to control odor concentration of both background stimulus and short pulses. 5 ml odor dilution in mineral oil was placed in a 20 ml glass bottles. Different concentrations were created by changing the air stream throught the vial and set to the value reported in the figures).

## Pharmacology

Stocks of picrotoxin (SIGMA) were prepared in dimethyl sulfoxide (DMSO; 100 mM) and diluted down to 5 µM in ringer. An equal amount of DMSO dissolved in ringer was used as control. In each fly, the response to a random stimulus was first measured with ringer only, then PTX or the control was applied for 2 min, and the response was measured again.

Stock solutions of CGP54626 (25 mM in DMSO – Tocris #1088) and SKF97541 (50 mM in water – Tocris #0379) were diluted down in ringer (to 25 µM and 40 µM respectively) before experiments. For *Figure 5G and H*, the response was first measured with ringer only, then the drug or control ringer was applied for 8 min, and the response was measured again. To further control for stimulus fluctuations, in *Figure 5I and L* the random stimulus was presented four times to each fly with 5–8 min interval in between repetitions: two times in ringer and two times with treatment. We did not observe differences between the two repetitions.

## Image processing

All data were analyzed in MATLAB (Mathworks), except for calculation of the mean intensity of the regions of interest that was performed in Fiji. Images were aligned to a reference frame using a customized routine. The frame rate of acquisition was about 20 Hz, but for the temporal analysis, exact acquisition times were used. To do this, fluorescence intensity was resampled by interpolation at exact multiples of the actual mean frame rate. Mean basal fluorescence ($F_0$) was calculated by averaging the activity during the 4 s preceding stimulus onset. Calcium responses were quantified as relative changes in fluorescence ($F$-$F_0$)/$F_0$. No correction for bleaching was applied in calcium responses during the 2 min stimulation period because bleaching occurred only within the first second of illumination, which was discarded from the analysis. A linear bleaching effect was removed from Syp-pHTomato signals, which was estimated for each odor and was glomerulus and concentration independent (see *Figure 6—figure supplement 1*). Further analysis was performed on the mean ΔF/F averaged across animals.

## LN model and NLN model

We fitted an Linear-Non-linear (LN model) to the mean change in fluorescence as previously described (*Martelli et al., 2013*). The model assumes that the response $r(t)$ can be described by a temporal filter $k(t)$ and a static function $f$. It also assumes that the temporal filter is linear and that the response therefore depends linearly on the value of current and past stimuli and not on their higher order statistics. We estimated the linear filter from the measured data by reverse correlation (*Chichilnisky, 2001*) corrected for high-frequency noise by ridge regression. The filter obtained was normalized and convolved to the stimulus $s(t)$:$s_{pr}(t) = \int_{-\infty}^{t} s(t')k(t-t')dt'$. In *Figure 2* and in *Figure 9* we used the binary state of the valve multiplied by the nominal odor concentration as stimulus: $s(t) = Cv(t)$ (it was not possible to obtain PID measurements at all concentrations). The static function $f$ was then estimated by plotting the measured response $r(t)$ as a function of the projected stimulus $s_{pr}(t)$ (*Figure 2F*, gray dots) and fitting either a linear function $f_{NL} = as_{pr} + b$ or a Hill function $f_{NL} = 1/\left(1 + \left(\frac{H}{s_{pr}}\right)^n\right)$. The LN model cannot be fitted to the initial, transient response and, therefore, the first 35 s after stimulus onset were discarded. Two-thirds of the remaining data were used to fit the model. The last one-third of the measurement was used to estimate the goodness of the model as the ratio $NR = \sqrt{P_R/P_N}$ between the residuals of the fit $P_R = \left\langle (r(t) - f_{LN}(t))^2 \right\rangle_t$ and the noise in the response $P_N = \left\langle \left\langle (r_i(t) - r(t))^2 \right\rangle_i \right\rangle_t$ with $r(t) = \langle r_i(t) \rangle_i$. A value of $NR < 1$ indicates a model prediction within the noise in the data. All models shown here have a $NR < 1$ (*Figure 7—figure supplement 1*).

A similar procedure was used for the Non-linear-Linear-Non-Linear model in *Figure 2H and L*, except that the stimulus was first pass through the Hill function fitted to in *Figure 2G*.

Three parametric functions were fitted to the linear filters: a single exponential, $k_1(t) = A_1 e^{-\frac{t}{\tau_1}}$, a single exponential plus a constant $k_2(t) = A_1 e^{-\frac{t}{\tau_1}} + A_2$, and a double exponential $k_3(t) = A_1 e^{-\frac{t}{\tau_1}} + A_2 e^{-\frac{t}{\tau_2}}$. The Bayesian Information Criterion (BIC) was used to select the model that best-fit the linear filter. Assuming Gaussian distributed errors, we calculated for each

model $\mathrm{BIC} = n \ln(\langle\sigma^2\rangle) + \mathrm{k}\ln(n)$, where $\langle\sigma^2\rangle$ is the mean squared residual, $n$ is the number of data points, and $k$ is the number of parameters in the model. The model with the lowest $\mathrm{BIC}$ was selected.

### Analysis of combinatorial odor representations

For *Figure 8*, Principal Component Analysis (PCA) was performed in the five-dimensional glomerular space on the concatenated steady-state response to all tested concentrations (the transient initial response was discarded for this analysis). Each point corresponds to the linear combination of the response of five glomeruli at a certain time $t$. The coefficient of variation, $CV = \frac{\sigma}{m}$, was calculated from the standard deviation $\sigma$ and mean $m$ of the stimulus representation along the first principal component (as plotted in *Figure 9C-C'*). The $CV$ of the binary stimulus, the valve state, $v(t)$, is 0.9. Stimuli of different intensity can be defined as $s(t) = Cv(t)$; therefore, their CV is the same as for $v(t)$. In both ORNs and PNs the $CV$ at low concentrations ranges around 0.43 for both odors. This number can be explained by the fact that the response of the glomeruli results from the convolution of the stimulus $s(t)$ and a linear filter with characteristic timescale τ. The population response variance is expected to decrease with τ. Taking all the timescales of the ORNs and PNs linear filters calculated from GCaMP3 fluorescence (*Figure 7D and 7H*), we estimated a mean integration time across glomeruli and concentrations of $\langle\tau\rangle = 0.54\ s$. The expected $CV$ in *Figure 9D-D'* was therefore calculated as the $CV$ of the convolution of the stimulus $s(t)$ with an exponential filter of timescale $\langle\tau\rangle$. The use of a faster sensor (smaller τ) should result in larger values of the variance in both ORNs and PNs and a larger expected CV.

### Analysis of slow adaptation

In *Figure 4*, for each glomerulus, all single traces at all concentrations were pooled together, sorted by the maximum response in the first 12 s, and binned in 10 bins between zero and the maximum response. Trials within the same bin were averaged, and the maximum response $r_m$ within every single odor step in the random sequence was identified. The maximum response was fitted as a function of time $r_m(t)$ for each concentration by a linear relationship in ORNs, $r_m(t) = A + Bt$, and an exponential decay in PNs, $r_m(t) = A + Be^{-\frac{t}{\tau}}$. In *Figure 4C*, the initial ($r_m^0$) and adapted ($r_m^\infty$) responses were estimated from the fitted parameters. The relationship between initial and adapted response was fitted by a linear function $r_m^\infty = a + br_m^0$, and the degree of adaptation was defined as $d_{adap} = 1 - b$. A value equal to 0 indicates no adaptation and no change in activity, values between 0 and 1 indicate partial adaptation, and a value of 1 indicates complete adaptation.

## Acknowledgements

We are grateful to Alberto Bernacchia for valuable feedback on data analysis, Jan Clemens, Giovanni Galizia, Marion Silies and Paul Szyszka for helpful comments on the manuscript, Toshihide Hige and Carsten Duch for useful discussions, Jing Wang and the Bloomington Stock Center for providing fly strains, Harald Wohlfarth and Christin Albrecht for performing experiments, Sandeep Shrestha for help with spike sorting, Sabine Kreissl and Christoph Kleineidam for help with setting up electro-physiology and pharmacology, and Alja Lüdke and Jan Peters for technical assistance. This work was supported by a fellowship of the Alexander von Humboldt Foundation, funding from the University of Konstanz and by the Zukunftskolleg of the University of Konstanz to CM, by the University of Goettingen to AF and CM and by the German Research Council (SFB 889/B4) to AF.

## Additional information

### Funding

| Funder | Grant reference number | Author |
| --- | --- | --- |
| Alexander von Humboldt-Stiftung | Postdoctoral Fellowship | Carlotta Martelli |
| Deutsche Forschungsgemeinschaft | SFB 889/B4 | André Fiala |

| University of Konstanz | Carlotta Martelli |
| Zukunftskolleg of the University of Konstanz | Carlotta Martelli |
| Georg-August-Universität Göttingen | André Fiala<br>Carlotta Martelli |

The funders had no role in study design, data collection and interpretation, or the decision to submit the work for publication.

## Author contributions

Carlotta Martelli, Conceptualization, Resources, Data curation, Software, Formal analysis, Funding acquisition, Validation, Investigation, Visualization, Methodology, Writing—original draft, Project administration, Writing—review and editing; André Fiala, Resources, Supervision, Funding acquisition, Project administration, Methodology, Writing—review and editing

## Author ORCIDs

Carlotta Martelli (iD) https://orcid.org/0000-0002-5663-6580
André Fiala (iD) https://orcid.org/0000-0002-9745-5145

## Decision letter and Author response

Decision letter https://doi.org/10.7554/eLife.43735.023
Author response https://doi.org/10.7554/eLife.43735.024

# Additional files

## Supplementary files

• Transparent reporting form
DOI: https://doi.org/10.7554/eLife.43735.021

## Data availability

All data generated or analyzed during this study are included in the manuscript and supporting files.

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
