## [Decision Letter]

Thank you for sending your article entitled "Adaptation on multiple timescales in the olfactory pathway of *Drosophila*" for peer review at *eLife*. Your article has being evaluated by three peer reviewers, and the evaluation has being overseen by a Reviewing Editor and Ronald Calabrese as the Senior Editor.

Given the list of essential revisions, including new experiments, the editors and reviewers invite you to respond within the next two weeks with an action plan and timetable for the completion of the additional work. We plan to share your responses with the reviewers and then issue a binding recommendation.

We will look forward to hearing from you as soon as possible so that we can complete the evaluation.

Summary:

In this manuscript, Martelli and Fiala focus on olfactory neurons in *Drosophila* and characterize adaptation at multiple successive steps in the signal processing using imaging. They show that calcium signals in dendrites of second-order projection neurons (PNs) adapt to flickering stimuli whereas those in axons of presynaptic olfactory receptor neurons (ORNs) do not. Reporters of vesicular release expressed in ORNs exhibited adaptation at similar time scale as PN dendritic calcium signals suggesting that decrease in vesicular release is the mechanisms underlying adaptation. The reviewers agree that the strongest and most interesting result in the paper is the finding that the slow decay in PN response is likely due to changes in presynaptic release. The manuscript would be improved if it were re-organized to focus on the exciting PN results. The reviewers also raised two concerns that would need to be addressed experimentally as well as a number of textual clarifications, as outlined below.

Essential revisions:

1) The finding that ORNs show no evidence of adaptation at the level of calcium signals in ORN terminals is contrary to all previous studies of ORN sensory responses using electrophysiology. It is possible that all forms of ORN adaptation are undone by the spike-to-calcium transformation, however this needs to be explicitly demonstrated and explained, given the bold nature of the claim. Calibration/controls need to be performed to be sure that any form of adaptation is not simply lost to the nonlinearities and filtering properties of the indicator(s). Specifically, the authors should couple the imaging results in Figure 1L, where they compare the response to a pulse and the response to a pulse on an odor background, with electrophysiological recordings. They should (1) record ORN spikes with the same odor delivery conditions and show that spikes exhibit adaptation, and (2) deliver a series of odor pulses of different concentrations and record ORN spikes and also Ca^2+^ responses to the same stimuli to figure out where the indicator saturates. If the authors showed that ORN spiking shows canonical adaptation and that the GCaMP signal is not saturated, this would strengthen the findings on ORN calcium signals. In addition, the authors should discuss potential nonlinearities in their calcium indicator and how that impacts and limits interpretations, placing them in context with previous electrophysiological results.

2) The claim that PN adaptation primarily reflects synaptic depression should be supported by additional evidence. RNAi effects can be incomplete; therefore the authors should block GABA-B signaling using pharmacology (CGP 54626) and examine its effects on PN decay. This manipulation was previously shown to increase the rate of decay in PNs, consistent with the idea that presynaptic GABA-B receptors regulate synaptic depression (Olsen, 2008; Root, 2008). They could also try to activate GABA-B receptors using SKF 97541 (Root, 2008) which should reduce synaptic depression. Application of low concentrations of Cd^2+^ (Kazama, 2008) can also be used to manipulate synaptic depression. Combining these pharmacological manipulations with imaging of presynaptic release and PN decay would more firmly establish that synaptic depression is responsible for the observed decay.

3) The paper uses two conceptually distinct definitions for adaptation. In analyzing linear filters, the bilobed filters are called 'adaptation', since their step response returns towards baseline (Figure 1D). Later, in analyzing long timescale changes in PN responses, adaptation is a multiplicative gain change (Figure 3, and subsection “Slow Adaptation in Odor-Evoked Calcium Activity of PNs”). Still later, in the CV analysis (Figure 7), ORN calcium indicator responses appear to change gain multiplicatively. At a minimum, the authors should clarify and distinguish between these different response properties. Though previous analyses have called bilobed impulse responses 'adapting' (for instance Howard Berg's early measurements in bacteria), in my view a good operational definition of adaptation might be 'a change in the processing properties of a cell'. Under that definition, a linear filter shape alone would not be viewed as adapting.

4) The LN models that are central to this paper could be improved by choosing a good way to scale the filter. The x-axis in Figures 1G, 6B, 7BB' have arbitrary units, and can be scaled differently depending on filter and stimulus. I did not understand why this was labeled 'predicted linear response ΔF/F' throughout. If it was indeed the predicted linear response, then the best fit slope of f*s plotted against r should equal 1 (by definition). Previous work (Baccus and Meister) has scaled the filter so that the variance of f*s is equal to the variance of s. I believe this would provide a clearer view of things, but a consistent scaling must be chosen to make nonlinearity shapes comparable throughout. Without doing that, the nonlinearities are essentially not possible to compare.

In addition, the authors should make some attempt to deconvolve the contribution of the indicator to the linear filters as was done in Schnell, 2014. Otherwise the filter parameters they are reporting are mostly properties of the indicator (hence the difference between GCaMP3 and GCaMP6 filters).

5) The time constant of a negative component (τ_2_) of a linear filter fitted to PN calcium signals was very different between figures. About 1 s in Figure 1D (rather fast), about 10 s in Figure 4N (rather slow, and similar to ORN vesicular release), and about 1 s in Figure 6 (rather fast). Why is the time scale so different between experiments? In Figure 6, odor concentration was varied, but τ_2_ never reached close to 10 s. Therefore, the difference in odor concentration used cannot be the explanation.

Related to this, because the shape of the filter is heavily dependent on odor concentration, I think it is useful to describe this earlier. My suggestion is to bring Figure 6 just after Figure 2 that describes the results of similar experiments but for ORNs.

In addition to these essential revisions, there are a number of text changes that are noted in the individual reviews below. These are straightforward to complete.

*Reviewer #1:*

Neurons are capable of adapting to stimuli that fluctuate over time. In this manuscript, Martelli and Fiala focus on olfactory neurons in *Drosophila* and characterize adaptation at multiple successive steps in the signal processing using imaging. They show that calcium signals in dendrites of second-order projection neurons (PNs) adapt to flickering stimuli whereas those in axons of presynaptic olfactory receptor neurons (ORNs) do not. Reporters of vesicular release expressed in ORNs exhibited adaptation at similar time scale as PN dendritic calcium signals suggesting that decrease in vesicular release is the mechanisms underlying adaptation. Perhaps the most notable finding is that while ORNs exhibit adaptation in terms of spikes, they do not in terms of presynaptic calcium (The reason behind it remains unknown). Although these results represent some contribution to the field, I have several major concerns and suggestions.

1) The title of the manuscript is "Adaptation on multiple timescales" and the text refers to both fast and slow adaptation. However, the study mainly focuses on the slow component and the distinction between fast and slow adaptation was not clear.

2) Inconsistency in the time scale of a negative component. The time constant of a negative component (τ_2_) of a linear filter fitted to PN calcium signals was very different between figures. About 1 s in Figure 1D (rather fast), about 10 s in Figure 4N (rather slow, and similar to ORN vesicular release), and about 1 s in Figure 6 (rather fast). Why is the time scale so different between experiments? In Figure 6, odor concentration was varied, but τ_2_ never reached close to 10s. Therefore, the difference in odor concentration used cannot be the explanation.

Related to this, because the shape of the filter is heavily dependent on odor concentration, I think it is useful to describe this earlier. My suggestion is to bring Figure 6 just after Figure 2 that describes the results of similar experiments but for ORNs.

3) Although the authors use the term "combinatorial/population activity/representation" to describe the results in Figures 5 and 7, this is an overstatement. In Figure 5, for example, PC1 already captures 95 and 97% of response variance for methyl acetate and benzaldehyde, respectively. This reflects the fact that 3 out of 5 texted glomeruli show little response and the remaining two glomeruli show highly correlated activity. This means that the results of analyses won't change if PC1 was replaced with either DM1 or DM4. Similar argument applies to Figure 7. Because addition of multiple glomeruli does not add any information, text on population/combinatorial activity should be removed.

*Reviewer #2:*

Martelli and Fiala measured calcium signals in *Drosophila* ORNs, PNs, and LNs in response to time varying stochastic odorant signals. They fitted LN models to these stimulus-response traces to investigate the dynamics and adaptation of responses at the first olfactory synapse. Based on their analysis, they drew several conclusions: (1) ORN calcium dynamics are distinct from ORN firing rate properties (measured elsewhere), (2) PNs adapt on fast and slow timescales, and (3) the slow adaptation observed in PNs is due to presynaptic changes in vesicle release.

The analysis in support of point (3) is nicely done, but I believe there are some issues to be resolved with analysis supporting claims (1) and (2).

1) The paper uses two conceptually distinct definitions for adaptation. In analyzing linear filters, the bilobed filters are called 'adaptation', since their step response returns towards baseline (Figure 1D). Later, in analyzing long timescale changes in PN responses, adaptation is a multiplicative gain change (Figure 3, and subsection “Slow Adaptation in Odor-Evoked Calcium Activity of PNs”). Still later, in the CV analysis (Figure 7), ORN calcium indicator responses appear to change gain multiplicatively. At a minimum, the authors should clarify and distinguish between these different response properties. Though previous analyses have called bilobed impulse responses 'adapting' (for instance Howard Berg's early measurements in bacteria), in my view a good operational definition of adaptation might be 'a change in the processing properties of a cell'. Under that definition, a linear filter shape alone would not be viewed as adapting.

2) This paper makes several claims about ORN response properties in calcium that are quite different from those measured in ORN spikes or LFP. Given that the calcium indicator is a nonlinear read out of calcium concentration, the authors should do more to explain how these differences could arise, or explain what differences between experiments could account for them.

a) Recent papers have reported bi-phasic impulse responses in ORN firing rates and LFP measurements (Martelli et al., Gorur-Shandilya et al.). In Figure 1, the authors show that calcium signals are not biphasic. I found the discussion of this difference a little lacking – how could this difference come to be?

b) My reading of Figure 1L and related prose was that calcium in ORN axon terminals represents actual concentration and is not subject to the adaptation (or cross-adaptation) that has been reported in the spiking. This may well be true, but is there a proposed mechanism by which this could work? It seems hard to explain. This seems to conflict with ORN spiking measured in Martelli et al., Gorur-Shandilya et al., and Nagel et al., so some kind of proposal as to why seems required.

c) At least two recent papers have reported that ORNs show Weber-like gain scaling (Cao et al. and Gorur-Shandilya et al.). Here, Figures 2 and Figure 1—figure supplement 2 showed all filters normalized to the same scale, so it was hard to see any gain changes. Do these calcium filters support Weber-like gain scaling at the ORN axon? I could not tell. But based on Figure 7, it looks like not. Again, it's not clear to me how this could occur (i.e., undoing an upstream gain change seems mechanistically difficult), or whether I'm misunderstanding the results presented here.

3) In Figure 7, the authors report that the CV is conserved in PNs but not ORNs. Here, it also seems possible (or likely) that the calcium indicator is saturating, or that nonlinearities in the indicator are dominating the ORN results, especially given how different GC3 and GC6 look when directly compared with these stimuli. Overall, to draw strong conclusions, the authors should address issues related to potential nonlinearities in their calcium indicator and how that impacts and limits interpretations.

4) The LN models that are central to this paper could be improved by choosing a good way to scale the filter. The x-axis in Figures 1G, 6B, 7BB' have arbitrary units, and can be scaled differently depending on filter and stimulus. I did not understand why this was labeled 'predicted linear response ΔF/F' throughout. If it was indeed the predicted linear response, then the best fit slope of f*s plotted against r should equal 1 (by definition). Previous work (Baccus and Meister) has scaled the filter so that the variance of f*s is equal to the variance of s. I believe this would provide a clearer view of things, but a consistent scaling must be chosen to make nonlinearity shapes comparable throughout. Without doing that, the nonlinearities are essentially not possible to compare.

*Reviewer #3:*

This study uses calcium imaging to characterize changes in temporal encoding between olfactory receptor neurons and projection neurons of the *Drosophila* antennal lobe. Differences in encoding of both odor identity and timing between these two stages of olfactory processing have been extensively investigated using both electrophysiology (Kaissling, 1987; Bhandawat, 2007; Olsen, 2010; Kim, 2010, 2015; Nagel, 2011, 2015; Martelli, 2013; Cafaro, 2016; Gorur-Shandilya, 2017) and imaging (Root 2008) and have been summarized in several reviews (e.g. Wilson, 2013). Current data suggests that multiple stages of adaptation occur at the level of olfactory neuron transduction (Kaissling, Nagel), spike generation (Nagel, Martelli, Gorur-Shandilya), synaptic transmission between ORNs and PNs (Kazama, Nagel), and due to feedback from local interneurons (Olsen, Root, Nagel). The current study differs from previous ones in primarily using calcium imaging to explore questions of temporal encoding.

Contrary to previous findings, they find no evidence of adaptation in ORNs as measured (mostly) by GCaMP3 fluorescence in ORN terminals. This is surprising, as previous studies have found evidence for adaptation on both short (Martelli, Nagel) and long (Gorur-Shandilya, Nagel) timescales, as well as adaptation to the stimulus mean (Kaissling, Nagel, Gorur-Shandilya) and variance (Gorur-Shandilya). Some attempts are made to control for nonlinearities and temporal filtering by the calcium indicator, however, I think the authors still interpret their results too strongly given the limitations of these methods. The authors do not attempt to calibrate the indicators to determine their temporal resolution or dynamic range, nor do they correct for the contribution of the indicators to the observed filters and nonlinearities computationally. I think these concerns also apply to the conclusions about ORN vs. PN population encoding.

The study also looks at differences in encoding between ORNs and PNs. They find evidence for a slow decay timescale (~10s) in PNs that is not present in ORNs. A slow timescale of decay in PN responses has been observed previously by Nagel et al., 2015, and has been hypothesized to arise from vesicle depletion. Here the authors provide experimental evidence for this idea by recording both presynaptic calcium, presynaptic release, and postsynaptic calcium. They find that the decay is present in vesicle release measurements and postsynaptic calcium but not presynaptic calcium. This is a nice experiment and contributes to our understanding of adaptation processes in the olfactory system.

Finally, the authors find that dynamics are not altered by either pharmacological blockade of GABA-A receptors with picrotoxin or presynaptic expression of GABA-B RNAi, and conclude that inhibition plays no role in these dynamics. I think this conclusion is also a bit strong as RNAi blockade can be incomplete.

1) The finding that ORNs show no evidence of adaptation at the level of calcium signals in ORN terminals is contrary to all previous studies of ORN sensory responses using electrophysiology. It is possible that all forms of ORN adaptation are undone by the spike-to-calcium transformation, however I think this needs to be explicitly demonstrated and explained, given the bold nature of the claim. For example, the authors might use either odor or optogenetics to generate calibrated numbers of spikes in ORNS (as in Jeanne and Wilson, 2005) and then use indicators optimized for detecting single spikes (GCaMP7) to measure the spike-to-calcium transformation. Indicators can be calibrated and their dynamic range determined using the methods of Ryan and colleagues (e.g. Sankaranarayanan and Ryan, 2000). Temporal filtering by the indicator can be deconvolved as in Schnell, Fairhall, and Dickinson, 2014. Finally, a computational model showing how the spike-to-calcium transformation undoes ORN adaptation would put this finding in context and explain the paradoxical nature of the present result. In the absence of such approaches, I think the claims about ORN dynamics and population codes need to be tempered by strong caveats about the limits of the indicators.

2) The claim that PN adaptation primarily reflects synaptic depression should be supported by additional evidence. RNAi effects can be incomplete; therefore the authors should block GABA-B signaling using pharmacology (CGP 54626) and examine its effects on PN decay. This manipulation was previously shown to increase the rate of decay in PNs, consistent with the idea that presynaptic GABA-B receptors regulate synaptic depression (Olsen, 2008; Root, 2008). They could also try to activate GABA-B receptors using SKF 97541 (Root, 2008) which should reduce synaptic depression. Application of low concentrations of Cd^2+^ (Kazama, 2008) can also be used to manipulate synaptic depression. Combining these pharmacological manipulations with imaging of presynaptic release and PN decay would more firmly establish that synaptic depression is responsible for the observed decay.

[Editors' note: further revisions were requested prior to acceptance, as described below.]

Thank you for resubmitting your work entitled "Adaptation on multiple timescales in the olfactory pathway of *Drosophila*" for further consideration at *eLife*. Your revised article has been favorably evaluated by Ronald Calabrese as the Senior Editor, a Kristin Scott as the Reviewing Editor, and two reviewers.

The manuscript has been improved but there are a few issues that need to be addressed before acceptance, as outlined below:

1) The analysis in Figure 2, in which the authors attempt to deconvolve the effects of the indicator are much improved. However, some points are not clear in the figure. For example, are the deconvolved traces in Figure 2C from ORN or PN data? What do the filters in A and B look like after deconvolution?

I wonder if this figure and text might be clearer for readers if the issues with analyzing these kinds of data were stated up front. Two important points made in this figure are that (1) in order to gauge temporal responses from Ca imaging data one must correct for indicator kinetics, and (2) when examining nonlinearities in an olfactory system one should take into account the front-end Hill nonlinearity present in odor-receptor interactions. I think these are important points for the field, so I would argue for putting the analyses incorporating these up top, rather than dwelling on the shape and predictive ability of the filters prior to deconvolution.

The authors have now done a much more thorough job of looking at the role of the indicator nonlinearities and filtering on their measurements of temporal processing. I think this deserves some mention in the Discussion, as the field often takes these at face value (e.g. Si et al., 2019).

2) The data comparing ORN firing rate responses and Ca responses to pulses on backgrounds are truly fascinating. In Figure 3F, I think it would be helpful to show an overlay of the actual Ca response on top of predicted Ca responses to see how far off these are. As a future direction, Ca influx into terminals is known to be highly nonlinear. It would be interesting to try to model this nonlinearity together with the sensor kinetics to see if the discrepancy between firing rate and Ca can be explained.

3) With regards to the inhibition pharmacology, I think the text could be clearer about the relationship between inhibition and depression proposed in prior literature. The work in Root, 2008 and Olsen, 2008, argues that synaptic depression is downstream of presynaptic inhibition: because presynaptic inhibition decreases presynaptic Ca^2+^, it should also decrease synaptic depression, leading to more facilitating synapses (this is shown in a supplement in the Olsen, 2008 paper). Overall the pharmacology shown in Figure 5 is rather at odds with that model. CGP should increase presynaptic calcium and increase PN depression, but the example shown shows no effect of CGP. (It might be helpful to show one of the glomeruli like VM2 where there was an increase in response with CGP). SFK should decrease presynaptic calcium and make PN responses more facilitating, but instead it seems to do the opposite! This is surprising and deserves some comment in the Discussion.

4) Authors clarified why old scaling looked as it did, and I think the new scaling of binary*dilution is more appropriate and gives sensible results. This also clarifies that the authors do see gain changes not too far from Weber scaling. However, in Figure 9 I think the authors went back to using the old scaling, without including the dilution. I think the authors should change this to binary*dilution for clarity and consistency.

---

## [Author Response]

[Editors' note: the authors’ plan for revisions was approved and the authors made a formal revised submission.]

Essential revisions:1) The finding that ORNs show no evidence of adaptation at the level of calcium signals in ORN terminals is contrary to all previous studies of ORN sensory responses using electrophysiology. It is possible that all forms of ORN adaptation are undone by the spike-to-calcium transformation, however this needs to be explicitly demonstrated and explained, given the bold nature of the claim. Calibration/controls need to be performed to be sure that any form of adaptation is not simply lost to the nonlinearities and filtering properties of the indicator(s). Specifically, the authors should couple the imaging results in Figure 1L, where they compare the response to a pulse and the response to a pulse on an odor background, with electrophysiological recordings. They should (1) record ORN spikes with the same odor delivery conditions and show that spikes exhibit adaptation, and (2) deliver a series of odor pulses of different concentrations and record ORN spikes and also Ca^2+^ responses to the same stimuli to figure out where the indicator saturates. If the authors showed that ORN spiking shows canonical adaptation and that the GCaMP signal is not saturated, this would strengthen the findings on ORN calcium signals. In addition, the authors should discuss potential nonlinearities in their calcium indicator and how that impacts and limits interpretations, placing them in context with previous electrophysiological results.

We agree with the reviewers that a direct comparison between firing rate and calcium is necessary to clarify and validate our results. Following the reviewers’ suggestions, we have performed single sensillum recordings and calcium imaging using the same odor stimuli. The results are presented in Figure 3 and Figure 3—figure supplement 1. We show a calibration curve for two ORNs, showing that the measured calcium responses fall in the exact same stimulus range as the firing rate response (Figure 3H and Figure 3—figure supplement 1C). Background experiments performed with different values of the background and test pulse, demonstrate the existence of a striking difference between presynaptic calcium and firing rate in the ORNs. The experiments were performed using GCaMP3 for consistency with the rest of the manuscript. However, in Figure 3—figure supplement 1D-E we also show results obtained with GCaMP6f. In this set of experiments, we have used a higher concentration of the odor to be able to measure the stimuli and to quantify the response of an additional glomerulus, VM7d. Moreover, we show that the calcium dynamics cannot be explained simply by the slow kinetics of the sensors (Figure 3F) and we discuss how non-linearities in the calcium indicator may affect the interpretation of the results.

2) The claim that PN adaptation primarily reflects synaptic depression should be supported by additional evidence. RNAi effects can be incomplete; therefore the authors should block GABA-B signaling using pharmacology (CGP 54626) and examine its effects on PN decay. This manipulation was previously shown to increase the rate of decay in PNs, consistent with the idea that presynaptic GABA-B receptors regulate synaptic depression (Olsen, 2008; Root 2008). They could also try to activate GABA-B receptors using SKF 97541 (Root, 2008) which should reduce synaptic depression. Application of low concentrations of Cd^2+^ (Kazama, 2008) can also be used to manipulate synaptic depression. Combining these pharmacological manipulations with imaging of presynaptic release and PN decay would more firmly establish that synaptic depression is responsible for the observed decay.

To address these concerns, we have performed additional pharmacology and present the results in Figure 5H-L. For this set of experiments, we decided to focus on glomerulus DM1, as this is the one we show in most figures. DM1 did not seem to receive much lateral inhibition with the odor stimulus used here, a fact that per se supports our hypothesis that lateral inhibition is not necessary for the slow PNs adaptation. Previous studies, mentioned above by the reviewer, used single odor pulses to test the effect of GABA-B agonist and antagonist. Therefore, in order to demonstrate that the pharmacology is working, we have conducted a control experiment where we present 3 odor pulses: methyl acetate, ethyl butyrate and the mixture of the two odors, and imaged the response of DM1 and DM4 (that only respond to methyl acetate) and VM2 (that only respond to ethyl butyrate). Presenting the mixture, we hoped to increase the lateral inhibition on DM1. Unfortunately, this was not the case and the response in all 3 glomeruli was the same with the corresponding single ligand or the mixture. Responses to the mixture were not reported in the manuscript as of marginal relevance for the paper, although they do constitute another evidence that lateral inputs are glomerulus and stimulus specific.

Tonic inhibition elicited by bath application of the GABA-B agonist, disrupts the calcium response of both ORNs and PNs. One important consequence of this approach is that forced inhibition is not stimulus dependent, as usually in normal conditions (also see Nagel et al., 2015). The expected outcome of this experiment was to decrease presynaptic activity and observe changes in postsynaptic response, however the results we obtained suggest a more complicated scenario that probably involves recursive activity in the AL network, understanding which is beyond the scope of the current manuscript. We have discussed these points in the subsection “The role of lateral inhibition in PN slow adaptation”.

3) The paper uses two conceptually distinct definitions for adaptation. In analyzing linear filters, the bilobed filters are called 'adaptation', since their step response returns towards baseline (Figure 1D). Later, in analyzing long timescale changes in PN responses, adaptation is a multiplicative gain change (Figure 3, and subsection “Slow Adaptation in Odor-Evoked Calcium Activity of PNs”). Still later, in the CV analysis (Figure 7), ORN calcium indicator responses appear to change gain multiplicatively. At a minimum, the authors should clarify and distinguish between these different response properties. Though previous analyses have called bilobed impulse responses 'adapting' (for instance Howard Berg's early measurements in bacteria), in my view a good operational definition of adaptation might be 'a change in the processing properties of a cell'. Under that definition, a linear filter shape alone would not be viewed as adapting.

We acknowledge the confounding use of the word ‘adaptation’ and we agree on making it clearer.

As the reviewer mentioned, the biphasic filter indicates that in presence of a step stimulus the response will have a phasic component, after which it comes back towards baseline and stays at an adapted value (Author response image 1 green). This happens at all intensities, although the decrease might not be the same for all stimulus intensities. If for example, the decrease is proportional to peak response (~50% in the example, Author response image 1), then adaptation will result in what we have defined before as a “multiplicative gain adjustment” (Author response image 1). This happens when the relative size of the two lobes of the filter is independent of stimulus intensity. However, no matter how the filter shape depends on concentration, there will be always a gain adjustment (not necessarily multiplicative). We agree that this naming is confusing because this is not in the strict sense a change of “response gain”, but rather a “linear rescaling of the representation”. We have clarified this in the subsection “Slow adaptation linearly rescales the combinatorial odor representation in PNs”.

Importantly, this change (multiplicative or not) is associate to ‘a change in the processing properties of a cell’ and to what should be properly called ‘gain adjustment’. Adaptation drives the response to a lower value (Author response image 1, stimulus 1). This means that the same stimulus induces a different response depending on whether the neuron is adapted to a background stimulus (cyan) or not (blue). We don’t know much about what happens if we apply a stimulus a little higher than the adapting one (stimulus 2). However, we can expect for continuity that there will be a response a little bit higher than the one to stimulus 1 (cyan), and certainly not as high as the response expected for stimulus 2 without a background (shaded blue). Therefore, adaptation has driven the response function in a new regime (dotted line), which we obtained, in this example, starting from the biphasic filter. The filter does not tell much about how the ‘response gain’ has changed (in the sense of Weber’s law), but its biphasic shape indicates that the neuron responds to *changes* in stimulus intensity, rather than the absolute stimulus intensity. The definition of adaptation we used in the paper are therefore not conceptually different, but related to each other: 1) a linear rescaling of the representation can result from a biphasic filter with specific concentration invariant properties, 2) a biphasic filter always induces a change of some kind in the representation and 3) for continuity, a change in the neuron response properties. We have connected a bit better these concepts in the text where appropriate.

Perhaps most of the confusion comes from the analysis conducted to capture the change in activity on long timescales, which cannot be explained by the fast filter obtained with the LN model. In order to provide a quantification of the slower adaptive component, we designed the analysis shown in Figure 4. This multiplicative rescaling can be likely reconducted to a biphasic filter with a much slower timescale, which we have extracted from the pH sensor measurements. We hope that the new arrangement of the paper makes these points clearer for the reader.

4) The LN models that are central to this paper could be improved by choosing a good way to scale the filter. The x-axis in Figures 1G, 6B, 7BB' have arbitrary units, and can be scaled differently depending on filter and stimulus. I did not understand why this was labeled 'predicted linear response ΔF/F' throughout. If it was indeed the predicted linear response, then the best fit slope of f*s plotted against r should equal 1 (by definition). Previous work (Baccus and Meister) has scaled the filter so that the variance of f*s is equal to the variance of s. I believe this would provide a clearer view of things, but a consistent scaling must be chosen to make nonlinearity shapes comparable throughout. Without doing that, the nonlinearities are essentially not possible to compare.

We apologize for the confusing use of the “ΔF/F” units assigned to the x axis of the old Figure 1G, 6B, 7BB'. These plots do show the measured response r as a function of f*s, as this is by definition the way we use to quantify the non-linear static function (Figure 2 and 4 of Chichilnisky, 2001). We have called this ‘predicted linear response’ because, apart for a multiplicative factor, this is the response of a purely linear system described by the filter f. However, since we normalize the filters by its amplitude, it is more appropriate to assign arbitrary units to it. We have changed the name of the axis to ‘projected stimulus’, similarly to Figure 1C-D of Gorur-Shandilya, 2017 (but with arbitrary units given the normalization we made).

The reviewers are also correct that an appropriate way to scale the filters is that of equalizing the variance of f*s to the stimulus variance, so that changes in sensitivity would appear only in the shape of the nonlinearity. This was our goal as well. Please note that our LN model was built using the binary signal from the valve as input stimulus (*same for all stimulus intensities*). We did so because we don’t have an actual measure of the stimulus for most dilutions used (we chose these dilutions on a logarithmic scale from 10^-6^ to 10^-2^ to cover the dynamic range of the response and the PID is unfortunately not sensitive in this full range).

When using the binary stimulus, a normalization of the filter by amplitude is essentially equivalent to a normalization of the filter by the response variance and results in f*s having variance ~1 (Author response image 2).

**Author response image 2. respfig2:** 

In order to address the reviewers’ concern, we now estimate the stimulus as the binary valve state multiplied by the dilution. When doing so, the filters are the same as before, but the non-linear function shows a decrease of response gain for increasing concentrations (Author response image 2 and main Figure 2F). In particular the gain scales with an exponent of ~ -0.75 (Author response image 2). In Gorur-Shandilya, 2017, this scaling factor was used to test whether the ORN response follows Weber’s law in response to gaussian stimuli with different mean (there the scaling exponent was ~ -1). The same paper has also shown that when using stimuli that shut down to zero (as binary stimuli or naturalistic plumes) an upfront non-linearity should be taken into account. The response is in fact spanning a largely non-linear regime and the gain change that we observe in the non-linear function is probably mostly related to the local gain of the dose response curve. To illustrate that, we have plotted the peak response (measured for each pulse in the random stimulus) as a function of the dilution (Author response image 2) and fitted a Hill function (black line). From this fit we have calculated the local gain around the 7 dilutions and plotted it as a function of the dilution (Author response image 2). This local gain decreases with an exponent of ~0.8 similarly to what found from the LN model analysis (Author response image 2). This suggest that the non-linear function obtained with a binary stimulus is capturing the nonlinearities of the dose-response curve.

Therefore, we have implemented an NLN model (Non-linear-Linear-Non-linear model) that takes this upfront non-linearity in account. The results are shown in Figure 2G and H.In addition to our previous observations (that response becomes non-linear and tends to saturate at higher concentrations), we now can nicely show that the response function at all stimulus intensities fall on a single curve. This demonstrate that the response properties are similar for a large range of concentrations. We thank the reviewers for pointing us this direction, that we have not initially considered.

In addition, the authors should make some attempt to deconvolve the contribution of the indicator to the linear filters as was done in Schnell, 2014. Otherwise the filter parameters they are reporting are mostly properties of the indicator (hence the difference between GCaMP3 and GCaMP6 filters).

As suggested by the reviewer, we have deconvolved the indicator kinetics from our measurements, as in Schnell, 2014. Doing so we could reveal a fast phasic component in the calcium response which is likely inherited from fast adaptation in the firing rate. We show in the manuscript how our conclusions drawn from the analysis of GCaMPs signal are or are not affected by the sensor kinetics. Thank you for pointing us to Schnell’s paper. It remains clear that there could be non-linearities in the sensor that are not taken in account by this approach.

5) The time constant of a negative component (τ_2_) of a linear filter fitted to PN calcium signals was very different between figures. About 1 s in Figure 1D (rather fast), about 10 s in Figure 4N (rather slow, and similar to ORN vesicular release), and about 1 s in Figure 6 (rather fast). Why is the time scale so different between experiments? In Figure 6, odor concentration was varied, but τ_2_ never reached close to 10 s. Therefore, the difference in odor concentration used cannot be the explanation.

We apologize that these results were not presented in a clear way. In the old Figure 1D and in Figure 6τ_2_ is the timescales obtained from fitting a double exponential function to the linear filter. But in Figure 4Nτinstead reported was obtained has shown in Figure 3A-D.

We have now rearranged the order of the figures and we hope that this confusion is eliminated. PNs dynamics have multiple timescales, a slow decrease in response over seconds (Figure 4) and fast phasic dynamics over hundreds of milliseconds (Figure 7).

Related to this, because the shape of the filter is heavily dependent on odor concentration, I think it is useful to describe this earlier. My suggestion is to bring Figure 6 just after Figure 2 that describes the results of similar experiments but for ORNs.

We thank the reviewers for this suggestion. We have now changed the figures and compared directly the linear filters for ORNs and PNs in Figure 7.

In addition to these essential revisions, there are a number of text changes that are noted in the individual reviews below. These are straightforward to complete.Reviewer #1:[…] 1) The title of the manuscript is "Adaptation on multiple timescales" and the text refers to both fast and slow adaptation. However, the study mainly focuses on the slow component and the distinction between fast and slow adaptation was not clear.

Fast adaptive components in ORNs and PNs response were previously studied and we here we aimed to show when and how they can be observed in calcium signals (Figures 2 and 7). We think that it is important to acknowledge the existence of multiple timescale and discuss their different functional roles.

2) Inconsistency in the time scale of a negative component. The time constant of a negative component (τ_2_) of a linear filter fitted to PN calcium signals was very different between figures. About 1 s in Figure 1D (rather fast), about 10 s in Figure 4N (rather slow, and similar to ORN vesicular release), and about 1 s in Figure 6 (rather fast). Why is the time scale so different between experiments? In Figure 6, odor concentration was varied, but τ_2_ never reached close to 10s. Therefore, the difference in odor concentration used cannot be the explanation.Related to this, because the shape of the filter is heavily dependent on odor concentration, I think it is useful to describe this earlier. My suggestion is to bring Figure 6 just after Figure 2 that describes the results of similar experiments but for ORNs.

We addressed these comments above.

3) Although the authors use the term "combinatorial/population activity/representation" to describe the results in Figures 5 and 7, this is an overstatement. In Figure 5, for example, PC1 already captures 95 and 97% of response variance for methyl acetate and benzaldehyde, respectively. This reflects the fact that 3 out of 5 texted glomeruli show little response and the remaining two glomeruli show highly correlated activity. This means that the results of analyses won't change if PC1 was replaced with either DM1 or DM4. Similar argument applies to Figure 7. Because addition of multiple glomeruli does not add any information, text on population/combinatorial activity should be removed.

The reviewer is right in saying that for methyl acetate the population response is dominated by 2 glomeruli, as quantified by the scores for PC1 that we have now reported in Figure 8—figure supplement 1. This odor induces responses in other glomeruli only at higher concentrations, but at these concentrations the new glomeruli do add information as DM4 and DM1 get saturated and do not follow the stimulus dynamics well. Moreover when using benzaldehyde, the contribution of the different glomeruli is more balanced, both in ORNs and even more in PNs. Since for both odors we obtained similar results, we decided to keep the population analysis.

Reviewer #2:[…] 1) The paper uses two conceptually distinct definitions for adaptation. In analyzing linear filters, the bilobed filters are called 'adaptation', since their step response returns towards baseline (Figure 1D). Later, in analyzing long timescale changes in PN responses, adaptation is a multiplicative gain change (Figure 3, and subsection “Slow Adaptation in Odor-Evoked Calcium Activity of PNs”). Still later, in the CV analysis (Figure 7), ORN calcium indicator responses appear to change gain multiplicatively. At a minimum, the authors should clarify and distinguish between these different response properties. Though previous analyses have called bilobed impulse responses 'adapting' (for instance Howard Berg's early measurements in bacteria), in my view a good operational definition of adaptation might be 'a change in the processing properties of a cell'. Under that definition, a linear filter shape alone would not be viewed as adapting.

We addressed this concern above.

2) This paper makes several claims about ORN response properties in calcium that are quite different from those measured in ORN spikes or LFP. Given that the calcium indicator is a nonlinear read out of calcium concentration, the authors should do more to explain how these differences could arise, or explain what differences between experiments could account for them.a) Recent papers have reported bi-phasic impulse responses in ORN firing rates and LFP measurements (Martelli et al., Gorur-Shandilya et al.). In Figure 1, the authors show that calcium signals are not biphasic. I found the discussion of this difference a little lacking – how could this difference come to be?

We have now added an analysis of the calcium responses deconvolved of the sensor kinetics, showing a fast phasic transient and a biphasic filter in the calcium response (Figure 2). We also show that when the filter shape is used as internal comparison (between ORNs and ORNs and PNs), the same conclusions could be drawn with or without deconvolution of the sensor.

b) My reading of Figure 1I and related prose was that calcium in ORN axon terminals represents actual concentration and is not subject to the adaptation (or cross-adaptation) that has been reported in the spiking. This may well be true, but is there a proposed mechanism by which this could work? It seems hard to explain. This seems to conflict with ORN spiking measured in Martelli et al., Gorur-Shandilya et al., and Nagel et al., so some kind of proposal as to why seems required.

The new experiments presented in Figure 3 aim to clarify this point. We did not investigate the mechanisms underlying background adaptation in ORN presynapses, but we discuss several possibilities in the Discussion session (subsection “Firing versus calcium adaptation in ORNs”).

c) At least two recent papers have reported that ORNs show Weber-like gain scaling (Cao et al. and Gorur-Shandilya et al.). Here, Figures 2 and Figure 1—figure supplement 2 showed all filters normalized to the same scale, so it was hard to see any gain changes. Do these calcium filters support Weber-like gain scaling at the ORN axon? I could not tell. But based on Figure 7, it looks like not. Again, it's not clear to me how this could occur (i.e., undoing an upstream gain change seems mechanistically difficult), or whether I'm misunderstanding the results presented here.

Please refer above to our answer to the major revisions.

3) In Figure 7, the authors report that the CV is conserved in PNs but not ORNs. Here, it also seems possible (or likely) that the calcium indicator is saturating, or that nonlinearities in the indicator are dominating the ORN results, especially given how different GC3 and GC6 look when directly compared with these stimuli. Overall, to draw strong conclusions, the authors should address issues related to potential nonlinearities in their calcium indicator and how that impacts and limits interpretations.

Please refer to the major revisions above. Saturation of the sensor was tested in Figure 2.

Of course, it should be noted that the deconvolution we performed is not cell specific and therefore internal comparison between ORNs and PNs remained valid (Figure 7). One possible cell specific non-linearity could arise from the expression level of the sensor in ORNs and PNs. In control experiments we have tested heterozygous and homozygous orcoGAL4>GCaMP6f flies (not in the manuscript). We found that the basal fluorescence was much lower in heterozygous flies, but the response dynamics were similar, suggesting that we are in a safe regime of the sensor concentration. Similarly, experiments on flies with one or two GCaMP3 constructs (e.g. Figure 5C and 5E), or with Syp/Homer-GCaMP3 (Figure 16) had different basal fluorescence but no differences in the response dynamics.

4) The LN models that are central to this paper could be improved by choosing a good way to scale the filter. The x-axis in Figures 1G, 6B, 7BB' have arbitrary units, and can be scaled differently depending on filter and stimulus. I did not understand why this was labeled 'predicted linear response ΔF/F' throughout. If it was indeed the predicted linear response, then the best fit slope of f*s plotted against r should equal 1 (by definition). Previous work (Baccus and Meister) has scaled the filter so that the variance of f*s is equal to the variance of s. I believe this would provide a clearer view of things, but a consistent scaling must be chosen to make nonlinearity shapes comparable throughout. Without doing that, the nonlinearities are essentially not possible to compare.

We addressed this concern above.

Reviewer #3:[…] 1) The finding that ORNs show no evidence of adaptation at the level of calcium signals in ORN terminals is contrary to all previous studies of ORN sensory responses using electrophysiology. It is possible that all forms of ORN adaptation are undone by the spike-to-calcium transformation, however I think this needs to be explicitly demonstrated and explained, given the bold nature of the claim. For example, the authors might use either odor or optogenetics to generate calibrated numbers of spikes in ORNS (as in Jeanne and Wilson, 2005) and then use indicators optimized for detecting single spikes (GCaMP7) to measure the spike-to-calcium transformation. Indicators can be calibrated and their dynamic range determined using the methods of Ryan and colleagues (e.g. Sankaranarayanan and Ryan, 2000). Temporal filtering by the indicator can be deconvolved as in Schnell, Fairhall, and Dickinson, 2014. Finally, a computational model showing how the spike-to-calcium transformation undoes ORN adaptation would put this finding in context and explain the paradoxical nature of the present result. In the absence of such approaches, I think the claims about ORN dynamics and population codes need to be tempered by strong caveats about the limits of the indicators.

Please find above an answer to this concern, that we have addressed using a deconvolution analysis. As a note, our current genetic approach and imaging set up do not allow for the single spike deconvolution suggested, which would require simultaneous imaging and electrophysiology in flies with single ORN labeled. For this paper we have used a driver line that expresses in orco positive neurons, and therefore, when we image from a single glomerulus, we are picking up signals from several ORNs expressing the same OR. We thank the reviewer for the suggested approach that could be used in the future, in combination with a refined labeling of single ORNs, to understand the mechanisms driving presynaptic calcium dynamics.

2) The claim that PN adaptation primarily reflects synaptic depression should be supported by additional evidence. RNAi effects can be incomplete; therefore the authors should block GABA-B signaling using pharmacology (CGP 54626) and examine its effects on PN decay. This manipulation was previously shown to increase the rate of decay in PNs, consistent with the idea that presynaptic GABA-B receptors regulate synaptic depression (Olsen, 2008; Root, 2008). They could also try to activate GABA-B receptors using SKF 97541 (Root, 2008) which should reduce synaptic depression. Application of low concentrations of Cd^2+^ (Kazama, 2008) can also be used to manipulate synaptic depression. Combining these pharmacological manipulations with imaging of presynaptic release and PN decay would more firmly establish that synaptic depression is responsible for the observed decay.

We addressed this concern above

[Editors' note: further revisions were requested prior to acceptance, as described below.]

The manuscript has been improved but there are a few issues that need to be addressed before acceptance, as outlined below:1) The analysis in Figure 2, in which the authors attempt to deconvolve the effects of the indicator are much improved. However, some points are not clear in the figure. For example, are the deconvolved traces in Figure 2C from ORN or PN data? What do the filters in A and B look like after deconvolution?I wonder if this figure and text might be clearer for readers if the issues with analyzing these kinds of data were stated up front. Two important points made in this figure are that (1) in order to gauge temporal responses from Ca imaging data one must correct for indicator kinetics, and (2) when examining nonlinearities in an olfactory system one should take into account the front end Hill nonlinearity present in odor-receptor interactions. I think these are important points for the field, so I would argue for putting the analyses incorporating these up top, rather than dwelling on the shape and predictive ability of the filters prior to deconvolution.The authors have now done a much more thorough job of looking at the role of the indicator nonlinearities and filtering on their measurements of temporal processing. I think this deserves some mention in the Discussion, as the field often takes these at face value (e.g. Si…Samuel, 2019).

Following these suggestions, we have edited the corresponding paragraph (“Dynamic properties of ORN calcium responses to binary stimuli”) to make clearer what the limits and problems are in analyzing calcium dynamics and responses to binary stimuli and how they can be overcome. The reviewer suggested to bring the deconvolution analysis up top before dwelling on the predictive ability of the linear model. However, we hope that the reviewers will agree that a direct look at the raw calcium dynamics must come first, as the deconvolution approach relies on a *model* of the sensor kinetics. We hope that it is now clear from the text in this paragraph, that the deconvolved model is a single one for all conditions and cell types, and therefore stimulus- and cell-specific differences described in this paper are preserved upon deconvolution. This is made clear in Figure 2E and 2I and in Figure 7B and 7F. Please note that we compare ORNs and PNs dynamics in Figure 7, and that Figure 7F shows that differences observed in the calcium reporter are preserved after deconvolution (similar to Figure 2A). The filters obtained with different sensors in Figure 2B look the same after deconvolution of the sensor kinetics (nor shown), as the deconvolved responses are very similar to each other (Figure 2C).

Finally, we have added a paragraph in the Discussion (subsection “Firing versus calcium adaptation in ORNs”) to recapitulate the approach and the conclusion we can draw from it.

2) The data comparing ORN firing rate responses and Ca responses to pulses on backgrounds are truly fascinating. In Figure 3F, I think it would be helpful to show an overlay of the actual Ca response on top of predicted Ca responses to see how far off these are. As a future direction, Ca influx into terminals is known to be highly nonlinear. It would be interesting to try to model this nonlinearity together with the sensor kinetics to see if the discrepancy between firing rate and Ca can be explained.

We have added the calcium trace in Figure 3F. We agree with the reviewer that non-linearities in the calcium influx could potentially explain the discrepancy. Further mechanistic insight should be gathered in the future by investigating the underlying molecular mechanisms, for example calcium channels expressed in these neurons and possible modulators of their activity. These points are mentioned in the last part of first paragraph of the Discussion (“Firing versus calcium adaptation in ORNs”).

3) With regards to the inhibition pharmacology, I think the text could be clearer about the relationship between inhibition and depression proposed in prior literature. The work in Root, 2008 and Olsen, 2008 argues that synaptic depression is downstream of presynaptic inhibition: because presynaptic inhibition decreases presynaptic Ca^2+^, it should also decrease synaptic depression, leading to more facilitating synapses (this is shown in a supplement in the Olsen, 2008 paper). Overall the pharmacology shown in Figure 5 is rather at odds with that model. CGP should increase presynaptic calcium and increase PN depression, but the example shown shows no effect of CGP. (It might be helpful to show one of the glomeruli like VM2 where there was an increase in response with CGP). SFK should decrease presynaptic calcium and make PN responses more facilitating, but instead it seems to do the opposite! This is surprising and deserves some comment in the Discussion.

The reviewer is right in saying that the interplay between depression and inhibition remains unclear from our data. Our goal with the pharmacological and genetic approach was to show whether the decrease in activity is a *direct* consequence of inhibition. Our data support that this is not the case, as also noted by a comment of the reviewer below. The data we have collected are not sufficient to demonstrate how the interplay works – unfortunately glomerulus VM2 was measured only in a set of experiments with single pulses and we don’t have data on the on the response to a random stimulus.

We totally agree with the view that depression is downstream of inhibition, but further work is necessary to understand how these two mechanisms couple under sustained stimulation and longer timescales. Indeed, a major important difference in Olsen, 2008 is the use of a minimal stimulation protocol that allows to measure single EPSC on timescales <100ms. In our experiments with SKF we are using relatively strong odor stimuli (they activate at least another glomerulus more strongly) delivered for a long time (>10sec). Our results are not in disagreement with the previous findings as they explore a much different response range (in time and network involvement). It is plausible that the effects we observe are due to interference with the dynamics of inhibitory inputs onto ORNs and possibly PNs, as well as on LNs themselves.

We have added these considerations in the Discussion (subsection “Synaptic depression as a mechanism for adaptation”).

4) Authors clarified why old scaling looked as it did, and I think the new scaling of binary*dilution is more appropriate and gives sensible results. This also clarifies that the authors do see gain changes not too far from Weber scaling. However, in Figure 9 I think the authors went back to using the old scaling, without including the dilution. I think the authors should change this to binary*dilution for clarity and consistency.

We have now replotted the non-linear functions in Figure 9B-B’ using the binary*dilution. Please note that these functions are not used for any of the analysis, but just show the stronger non-linear shape of PNs and their wider dynamic range.

Regarding Weber law, we would like to point out that the relationship between gain and stimulus in Figure 2J-inset is still quite far from Weber law, as Weber law would implicate an exponent of -1. As the results shown for the calcium in Figure 3 also don’t fit with Weber law, it cannot be excluded that the small dependency of response gain on stimulus amplitude (in Figure 2J-inset) could be due to a small error in the estimate of the front-end non-linearity. We hope that the reviewer agrees that a binary stimulus as the one used here is not appropriate to test for Weber law as done in Gorur-Shandilya et al., 2017, where stimulus mean and variance were separately controlled to measure response to small fluctuations around a varying mean concentration.